# Diffusible fraction of niche BMP ligand safeguards stem-cell differentiation

Sharif M. Ridwan [1,6], Autumn Twillie [1,6], Samaneh Poursaeid[1], Emma Kristine Beard[1], Muhammed Burak Bener[1], Matthew Antel [1], Ann E. Cowan[2,3], Shinya Matsuda [4,5] & Mayu Inaba [1]

*Drosophila* male germline stem cells (GSCs) reside at the tip of the testis and surround a cluster of niche cells. Decapentaplegic (Dpp) is one of the well-established ligands and has a major role in maintaining stem cells located in close proximity. However, the existence and the role of the diffusible fraction of Dpp outside of the niche have been unclear. Here, using genetically-encoded nanobodies called Morphotraps, we physically block Dpp diffusion without interfering with niche-stem cell signaling and find that a diffusible fraction of Dpp is required to ensure differentiation of GSC daughter cells, opposite of its role in maintenance of GSC in the niche. Our work provides an example in which a soluble niche ligand induces opposed cellular responses in stem cells versus in differentiating descendants to ensure spatial control of the niche. This may be a common mechanism to regulate tissue homeostasis.

The stem cell niche was initially proposed to be a limited space in tissues or organs where tissue stem cells reside. Based on the phenomenon in which transplantation of hematopoietic stem cells is only successful when naïve stem cells are depleted, a niche is thought to provide a suitable environment for stem cells to self-renew[1,2]. At the same time, the niche environment should not foster the differentiation of descendant cells to ensure that the correct balance of self-renewal and differentiation is maintained[2-4]. Although 40 years have passed since this niche concept was originally proposed[1], the mechanism of niche signal restriction is still poorly understood[5]. This is partly because of the difficulty in studying stem cells in their in vivo context. Moreover, assessment of the dispersion of soluble molecules in vivo is challenging.

The *Drosophila* male germline stem cell system provides a model to study niche-stem cell interactions. The testicular niche, called the hub, is composed of post-mitotic hub cells. Each testis contains a single hub harboring 8-14 germline stem cells (GSCs) that directly attach to the hub[6] (Fig. 1A). The division of a GSC is almost always asymmetric, via formation of a stereotypically oriented spindle, producing a new GSC and a gonialblast (GB), the differentiating daughter cell[7] (Fig. 1B). After being displaced away from the hub, the GB enters 4 rounds of transit-amplifying divisions to form 2- to 16- cell spermatogonia (SGs) (Fig. 1A). Then, 16-cell SGs become spermatocytes (SCs) and proceed to meiosis[8].

The Bone Morphogenetic Protein (BMP) ligand is often utilized in many stem cell niches in diverse systems[9]. In the *Drosophila* testis, the BMP ligand Decapentaplegic (Dpp) has emerged as a major ligand in the GSC niche together with a cytokine-like ligand, Unpaired (Upd)[10-14]. In the testis, it has been hypothesized that these ligands are mainly expressed in hub cells and activate signals in GSCs in close contact with the hub, and do not activate signals in GBs that are detached from the hub. However, it is unclear whether the fraction of ligands present beyond the niche space has a role, if it exists at all.

We previously demonstrated that hub-derived Dpp is received by GSC-specific membrane protrusions, which we termed microtubule based (MT)-nanotubes, to efficiently activate downstream pathways within the GSC population[15]. MT-nanotubes likely provide sufficient surface area along their length to allow the plasma membranes of GSCs and hub cells to closely contact one another for signaling[15,16]. This

[1]Department of Cell Biology, University of Connecticut Health Center, Farmington, CT, USA. [2]Richard D. Berlin Center for Cell Analysis and Modeling, University of Connecticut Health Center, Farmington, CT, USA. [3]Department of Molecular Biology and Biophysics, University of Connecticut Health Center, Farmington, CT, USA. [4]Biozentrum, University of Basel, Basel, Switzerland. [5]Present address: Department of Biological Sciences, Graduate School of Science, The University of Tokyo, Tokyo, Japan. [6]These authors contributed equally: Sharif M. Ridwan, Autumn Twillie. ✉e-mail: shinyamatsuda0423@gmail.com; inaba@uchc.edu

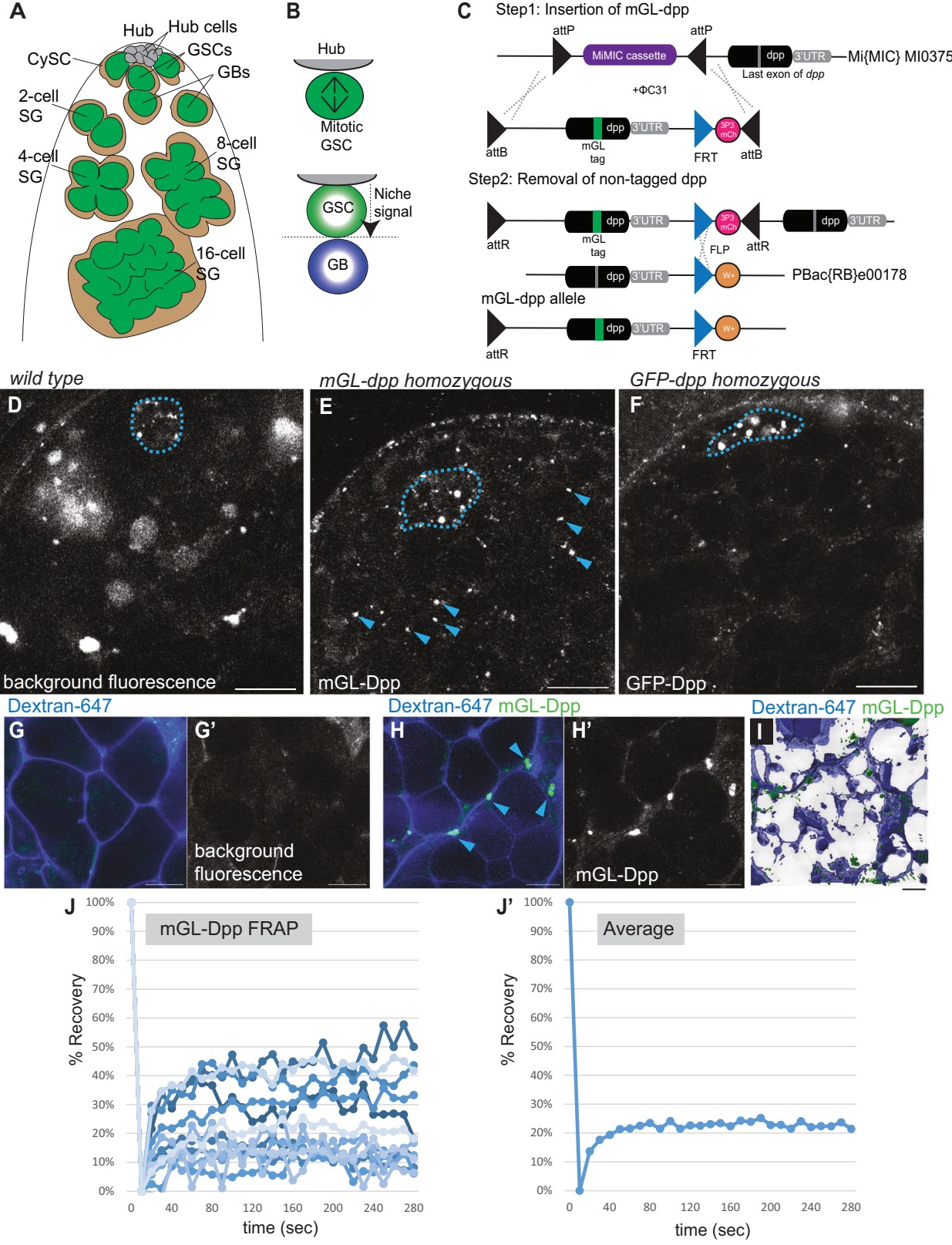

suggested the possibility that the Dpp signal is transmitted in a contact-dependent manner.

However, we also serendipitously found that Dpp overexpressed in the hub is freely diffusible[17], suggesting that Dpp could provide both contact-dependent and contact-independent signals. Besides the apparent contact-dependent signaling role of Dpp in the niche, we wondered what role (if any) the diffusing fraction of Dpp plays in the cells located outside of the niche.

In this study, we now directly address the function of the diffusible fraction of Dpp outside of the niche making use of a previously established tool, Morphotrap, a genetically encoded nanobody that can trap secretory ligands on the plasma membrane of ligand-

**Fig. 1 | Diffusible fraction of Dpp is present in the testis. A** Anatomy of anterior area of *Drosophila* testis. Hub cells form a cluster and serve as the niche for germline stem cells (GSCs). Differentiating daughter cells or gonialblasts (GBs) undergo four rounds of incomplete division, called spermatogonia (SGs). Somatic cyst stem cells (CySCs) or cyst cells (CCs) are encapsulating developing germline. **B** A schematic of asymmetric division (ACD) of GSCs. When the GSC divides, the mitotic spindle is always oriented perpendicularly towards hub-GSC interface (upper panel). As the result, GSC and GB are stereotypically positioned, one close to the hub and the other away from the hub (lower panel). Signal from the hub only activates the juxtaposed daughter cell so that the two daughter cells can acquire distinct cell fates. **C** A design of *mGreen Lantern (mGL)-dpp* allele. mGreen Lantern (mGL)-tagged *dpp* (last exon containing full-length *dpp* coding sequence after last processing site) with a 3xP3-mCherry cassette flanked by two inverted ΦC31 attB sites was replaced with gene trap cassette flanked by two inverted ΦC31 attP sites in

MiMICMI03752. Then endogenous *dpp* last exon was removed by recombination between FRT sites of pBac(RB)e00178 and *mGL-dpp* cassette. *w*; dominant white marker. See details in Methods. **D–F** Representative images comparing testis tips isolated from wildtype (*yw*, **D**), homozygous *mGL-dpp* fly (**E**) or homozygous *GFP-dpp* CRISPR knock-in fly (**F**) taken by the same microscope setting. **G–I** Representative images of spermatogonial cysts after incubation with Alexa Fluor 647 conjugated dextran-dye. Wildtype (*yw*, **G**) *mGL-dpp* fly (**H**) and the 3D rendering (using Imaris10.0) of a z-stack series of *mGL-dpp* (**I**). Arrowheads in (**E**) and **H** indicate mGL-Dpp puncta. **J, J'** Recovery curves of mGL-Dpp FRAP curves. % Recovery values (see Methods for calculation) from 13 trials are shown in (**J**). Values from each trial are shown in different colors. **J'** shows average values of 13 trials. Source data are provided in a Source Data file. All scale bars represent 10 μm. Asterisks indicate the approximate location of the hub. Live tissues were used for all images.

secreting cells[18,19]. Unexpectedly, we found that Dpp has roles in differentiating germ cells that are distinct and opposite of its actions in GSCs. In contrast to its GSC-specific function of promoting GSC maintenance, Dpp ensures differentiation of GB and SGs by blocking de-differentiation.

## Results

### Diffusible fraction of Dpp is present in the testis

We previously showed that overexpressed Dpp (expression of *UAS-dpp-mCherry*[20] under the hub-specific driver, *updGal4*, FBti0002638) can diffuse outside of the niche[17]. However, we were not able to visualize Dpp at endogenous levels because we could not detect signal outside of the hub above the background level by using two previously generated CRISPR knock-in lines, *GFP-dpp* (gift from Thomas Kornberg) and *mCherry-dpp*[17,21] (Table S1).

In this study, we tackled this challenge by generating a new line that expresses mGreen Lantern-tagged Dpp (*mGL-dpp*) from the endogenous *dpp* locus (Fig. 1C, Table S1)[22], so that we can visualize endogenous Dpp in the testis. Because the mGL-Dpp signal was not distinguishable from background fluorescence in the heterozygous flies, we attempted to obtain homozygous flies. However, we found that the homozygous *mGL-dpp* allele was semi-lethal, likely due to a defect in embryonic development. We therefore used a transgenic allele (*pPA dpp 8391/X*), containing the genomic region of *dpp*, that is only expressed in embryonic stages[23] to bypass the lethality by supplying the non-tagged Dpp in embryonic development. These rescued *mGL-dpp* homozygous flies were fully viable and able to reach adulthood without any developmental defects. Knock-down of mGL-tagged Dpp in the hub (*pPA dpp 8391/X, fasIII > GFP-IR, mGL-dpp/mGL-dpp*) significantly reduced in GSCs the level of phosphorylated Mad (mothers against dpp) protein (pMad), a readout of Dpp signal activation, confirming that mGL-Dpp is functional and that the rescuing transgene (*pPA dpp 8391/X*) is not expressed in the niche to a level that is able to rescue defects resulting from knocking down mGL-Dpp (Figure S1A), indicating that homozygous *mGL-dpp, pPA dpp 8391* line (referred to as *mGL-dpp with pPA dpp 8391* hereafter) can be a useful tool to examine the behavior of endogenous Dpp in the testis.

All tested lines (*mGL-dpp with pPA dpp 8391, GFP-dpp, mCherry-dpp, mScarlet-dpp with pPA dpp 8391*, Table S1) showed a punctate-pattern signal in the hub (Fig. 1D–F, Fig. S1B–E, Table S1), likely representing the combination of secreted Dpp, and potentially internalized Dpp-receptor complex, and background autofluorescence, often observed in the hub (Fig. 1D) as we reported previously[17]. In addition, we observed signal outside of the hub towards the SC regions in *mGL-dpp with pPA dpp 8391* testes (Fig. 1D–I). This result indicates that endogenous Dpp is likely diffusible from the niche in the fly testis.

As we reported previously, we could not detect signal outside of the hub above the background level in the two previously generated CRISPR knock-in lines, *GFP-dpp* and *mCherry-dpp* (Fig. S1B, C)[17]. The difference of distribution pattern among these lines may be explained

by different location of the fluorescence tag. In the previously generated CRISPR knock-in lines, tags were inserted upstream of the last processing site of Dpp, as compared to mGL-Dpp, where the tag is inserted after the last processing site of Dpp. Thus, there is a possibility that endogenous Dpp is not tagged efficiently in the *GFP-dpp* and *mCherry-dpp* lines (Table S1). Supporting this idea, *mScarlet (mSC)-dpp with pPA dpp 8391* (Table S1), in which mSC tag is inserted in the same location as the mGL in the *mGL-dpp* line[24], showed a similar distribution pattern to mGL-Dpp (Fig. S1E). Therefore, we conclude that our *mGL-dpp with pPA dpp 8391* line provides reliable readout of endogenous Dpp localization.

mGL-Dpp localized outside of the niche in a pattern reminiscent of the extracellular space between cells throughout the anterior regions of testis. We incubated testes isolated from homozygous *mGL-dpp with pPA dpp 8391* flies in media with freely diffusible fluorescent 10KDa dextran dye (Dextran-647), which is of a similar size to Dpp protein (14.8 kDa), for a short period of time (5-to-30 min). We observed that the Dextran-647 penetrates between interconnected germ cells at various SG stages, as described previously (Fig. 1G)[22]. We found that the fluorescence of mGL-Dpp co-localized with the area illuminated by Dextran-647 (Fig. 1H). Three-dimensional rendering of a z-stack further confirmed localization of mGL-Dpp in Dextran-647 dye-illuminated extracellular space (Fig. 1I). To assess whether the mGL-Dpp observed outside of the niche is a diffusible fraction of Dpp, we tested the mobility of mGL-Dpp using fluorescence recovery after photobleaching (FRAP). We first photo-bleached the mGL-Dpp signal in a small portion of the testis outside of the niche and monitored fluorescence recovery. After photobleaching, ~20% of mGL-Dpp signal on average recovered within a few minutes (Fig. 1J-J'), suggesting that a mobile, likely freely circulating fraction (20% of the total) of mGL-Dpp exists outside of the niche. This result also indicates that there is an immobile fraction (80% of the total) of mGL-Dpp that is likely bound to extracellular molecules/receptors or internalized by cells.

Taken together, these data demonstrate that a mobile fraction of Dpp is present not only in the niche but also in the rest of the tissue.

### Disturbing Dpp diffusion without affecting niche-GSC signal

Since differentiating cells are immediate descendants of stem cells, it is difficult to distinguish the direct and specific effects of niche ligands on differentiating cells. In fact, while Dpp function within the niche is well-characterized, the role for a potentially diffusible Dpp fraction outside of the niche is completely unknown. In order to assess the function of the diffusible fraction of Dpp, we sought to specifically disturb only the diffusible fraction of Dpp without affecting the niche-GSC Dpp signal. To achieve this, we utilized the Morphotrap, a genetically encoded tool consisting of a fusion protein between a transmembrane domain and a nanobody that recognizes green fluorescent protein (GFP) including its variants, such as mGL, and acts as a synthetic receptor for GFP-tagged proteins[18,19]. We used two versions of Morphotrap, each expressing a fusion protein of anti-GFP nanobody

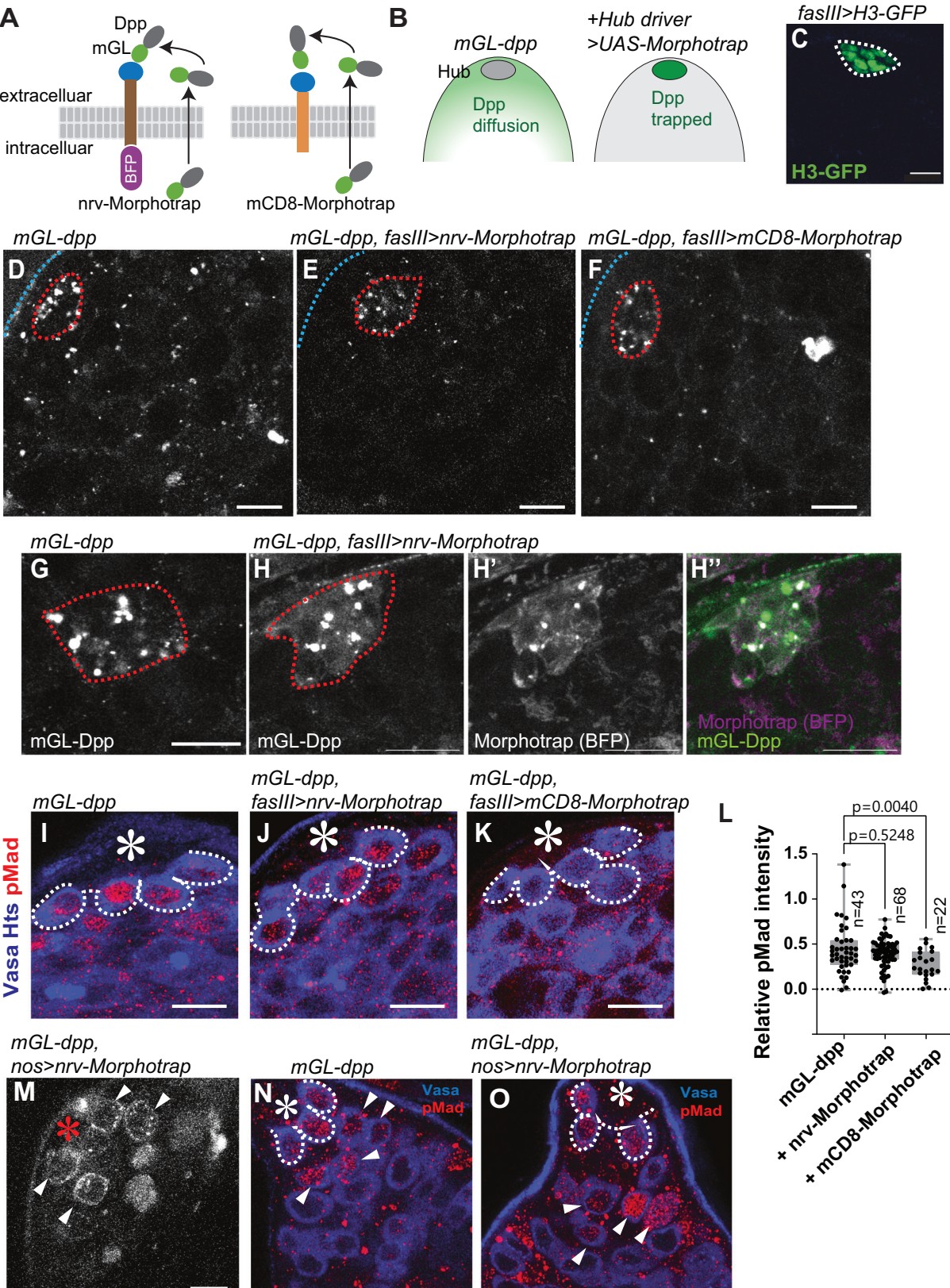

fused to one of two different transmembrane proteins, Nrv1 or mCD8 (Fig. 2A). Nrv-Morphotrap consists of the Nrv1 protein scaffold and localizes to the basolateral compartment of the plasma membrane[19]. mCD8-Morphotrap consists of the membrane protein mCD8 and localizes throughout the entire plasma membrane[19]. In order to trap the diffusible fraction of Dpp on the surface of niche cells, we utilized

the hub driver fasIIIGal4, which drives expression specifically in the hub cells that make up the germline stem cell niche (Fig. 2B, C). By expressing Morphotrap under control of the fasIIIGal4 driver in the *mGL-dpp with pPA dpp 8391* background, we reasoned that we could remove all circulating fractions of Dpp, including fractions secreted from the hub cells or from any other cell types, and trap it on hub cell

**Fig. 2 | Disturbing Dpp diffusion without affecting niche-GSC signal.**
**A** Schematics of the design to trap Dpp on the surface of hub cells using Morphotrap. The anti-GFP nanobody, vhhGFP4 (blue circle), is fused to transmembrane domains of either mouse CD8 (mCD8-Morphotrap) or Nrv1 basolateral scaffold protein (Nrv-Morphotrap). BFP; tagBFP fluorescent protein. **B** Expected outcome of hub-driven expression of Morphotrap in the background of *mGL-dpp* homozygous fly rescued by *pPA dpp 8391/X*. **C** Histone H3-GFP expressed under the hub-specific driver, fasIIIGal4. **D, F** Representative images of homozygous *mGL-dpp* fly rescued by *pPA dpp 8391/X* without (**D**) or with (**E**) fasIIIGal4 driven Nrv-Morphotrap expression or with (**F**) fasIIIGal4 driven mCD8-Morphotrap expression. Hub is encircled by red broken lines. **G, H** Hub area from homozygous *mGL-dpp* fly rescued by *pPA dpp 8391/X* without (**G**) or with (**H**) fasIIIGal4 driven Nrv-Morphotrap expression. **I–K** pMad staining of GSCs after trapping Dpp using indicated Morphotrap lines. The fasIIIGal4 driver was used. White broken lines encircle GSCs. **L** Quantification of pMad intensity in GSCs (relative to CCs) of fasIIIGal4 driven Nrv-Morphotrap or mCD8-Morphotrap expressing testes in *mGL-dpp* homozygous background rescued by *pPA dpp 8391/X*. P-values were calculated by Dunnett's multiple comparisons tests. n indicates the number of scored GSCs. Box plots show 25–75% (box), minimum to maximum (whiskers) with all data points. Source data are provided as a Source Data file. **M** Live testis tip of homozygous *mGL-dpp* fly rescued by *pPA dpp 8391/X* expressing Nrv-Morphotrap under the germline specific driver, nosGal4. Trapped mGL-Dpp signal is seen on the surface of early germ cells (white arrowheads). **N, O** pMad staining shows emerging pMad positive germ cells (arrowheads in **O**) outside of the niche of homozygous *mGL-dpp* fly rescued by *pPA dpp 8391/X*, without (**O**) or with (**N**) the expression of Nrv-Morphotrap under the germline specific driver, nosGal4. pMad positive germ cells are normally only seen in GSCs and immediate descendants around the hub (arrowheads in **N**). White broken lines encircle GSCs. All scale bars represent 10 µm. Asterisks indicate the hub. Live tissues were used for **C–H, M** and fixed samples were used for **I–K, N, O**.

membranes. This would thus prevent Dpp effect outside of the niche while keeping the contact-dependent signal from hub cells to stem cells intact (Fig. 2B).

As expected, we found that expression of both Nrv-Morphotrap and mCD8-Morphotrap under the fasIIIGal4 driver drastically reduced mGL-Dpp signal throughout the testis (Fig. 2D–F), indicating that both Morphotrap constructs can efficiently trap mGL-tagged Dpp. Trapped mGL-Dpp were detected in the hub colocalizing with BFP (blue fluorescent protein) fused to the intracellular domain of Nrv-Morphotrap (Fig. 2G, H"). However, although we were expecting to see a strong Dpp accumulation on the surface of hub cells, the signal was not very high. Therefore, we wondered if trapped mGL-Dpp may be constantly internalized and degraded in hub cells. If so, we would expect to see an increased mGL signal in hub lysosomes when lysosome digestion is perturbed by chloroquine (CQ) treatment, a drug that inhibits lysosomal enzymes. Indeed, we observed more mGL-Dpp positive punctae after 4-hour chloroquine treatment of Morphotrap-expressing testes, as compared to chloroquine-treated control *mGL-dpp with pPA dpp 8391* testes without Morphotrap expression (Fig. S2A–C), suggesting that trapped Dpp may be constantly degraded in hub cells.

Next, to examine whether Morphotrap expression specifically perturbs the diffusible fraction of Dpp and keeps the hub-GSC signal intact, we stained the testis for pMad to assess Dpp signal activation. In a normal testis, the level of phosphorylated Mad (pMad) is high in GSCs and immediately becomes lower in GB/SGs. We found that *mGL-dpp with pPA dpp 8391, fasIII>nrv-Morphotrap* testes showed similar pMad intensity in GSCs as compared to control *mGL-dpp with pPA dpp 8391* testes (Fig. 2I, J; quantified in 2 L). In contrast, pMad intensity was reduced in GSCs in *mGL-dpp with pPA dpp 8391, fasIII > mCD8-Morphotrap* testes as compared to control *mGL-dpp with pPA dpp 8391* testes (Fig. 2I, K; quantified in 2 L), indicating that with mCD8-Morphotrap, the hub-GSC signal may be affected. Therefore, we decided to use Nrv-Morphotrap out of concern that the mCD8-Morphotrap may have effects beyond disrupting the diffusible Dpp fraction. Trapping Dpp with either Morphotrap did not perturb pMad in somatic cyst cells (CCs) as compared to *mGL-dpp with pPA dpp 8391* control (Fig. S2D–F; quantified in S2G), which suggests that pMad in CCs is not activated by circulating Dpp and was used as an internal control for quantifying relative pMad intensity in germ cells for the following pMad quantifications (see *Methods* for more details).

In contrast to Morphotrap expression in hub cells, in which trapped mGL-Dpp is likely internalized and degraded, we found that expression of Nrv-Morphotrap using the germline driver nosGal4 (*mGL-dpp with pPA dpp 8391, nos>nrv-Morphotrap*) enabled us to visualize mGL-Dpp trapping along the membranes of germ cells outside of the niche (Fig. 2M). Consistent with this, we also observed hyper-activation of signaling as indicated by elevated pMad staining as compared to control *mGL-dpp with pPA dpp 8391* without Nrv-Morphotrap (Fig. 2N, O). These data confirm that a fraction of diffusible Dpp is present and trappable by the Morphotrap method, and that trapped Dpp can still signal to receptors present on the plasma membrane of the cells for contact-dependent function of Dpp.

Based on these results, we concluded that fasIII>Nrv-Morphotrap expression in the *mGL-dpp with pPA dpp 8391* background was the best tool to be used to assess the function(s) of a diffusible fraction of Dpp outside of the niche, without disrupting its function within the niche.

## The diffusible fraction of Dpp prevents de-differentiation

In the *Drosophila* testis, GSCs almost exclusively divide asymmetrically to produce one GSC and one GB (asymmetric outcome, Fig. 3A)[7,25,26]. However, symmetric outcomes (Fig. 3A) can be also caused[25,26] via two mechanisms: 1) spindle misorientation, where the mitotic spindle orients parallel to the hub-GSC interface, resulting in two GSCs (Fig. 3A–1)[7], and 2) de-differentiation, where a differentiating GB or SG physically relocates back to the niche and reverts to a GSC identity (Fig. 3A–2)[27]. Although a recent study suggested that the de-differentiation from the Bam-positive lineage (4-16-cell SGs) is required for maintenance of stem cell number only under challenging conditions[28], earlier lineage of GSC daughter cells (GB to 2-cell SGs) reenter the niche more frequently even under physiological conditions[26], indicating that this mechanism is critical for maintenance of the stem cell pool. At the same time, excessive de-differentiation has been hypothesized to be a cause of tumorigenesis[29].

By scoring the orientation of cells still interconnected by the fusome, a germline-specific organelle that branches throughout germ cells during division, we can quantify the frequency of symmetric outcomes in the niche[26,27,30]. We noticed that *mGL-dpp with pPA dpp 8391, fasIII>nrv-Morphotrap* testes showed a significantly higher frequency of symmetric events than control testes (Fig. 3B–D), suggesting that preventing Dpp diffusion results in more symmetric outcomes in GSC divisions.

In *mGL-dpp with pPA dpp 8391, fasIII>nrv-Morphotrap* testes, the number of GSCs at the hub was slightly higher at the timepoints of day 14 and 21 post-eclosion (Fig. 3E), suggesting the possibility that preventing Dpp diffusion may cause excess de-differentiation.

In the *Drosophila* testis, Bag of marbles (Bam), a translational repressor, is expressed after a germ cell exits the GSC state and is sufficient to promote differentiation[27]. Using heat-shock inducible expression of Bam in GSCs, we can deplete all GSCs in the niche. The niche is then replenished through de-differentiation once the flies are shifted back to normal temperature[27]. By introducing the *hs-bam* transgene in the *mGL-dpp with pPA dpp 8391/X* with or without fasIII>Nrv-Morphotrap expression, we assessed the function of the diffusible Dpp fraction on de-differentiation. Strikingly, *mGL-dpp with pPA dpp 8391/X, fasIII>nrv-Morphotrap* flies showed significantly faster recovery of GSCs in the niche after bam-induced differentiation, as compared to flies without fasIII>Nrv-Morphotrap expression (Fig. 3F–H). Moreover, Dpp trapping using an alternative driver/

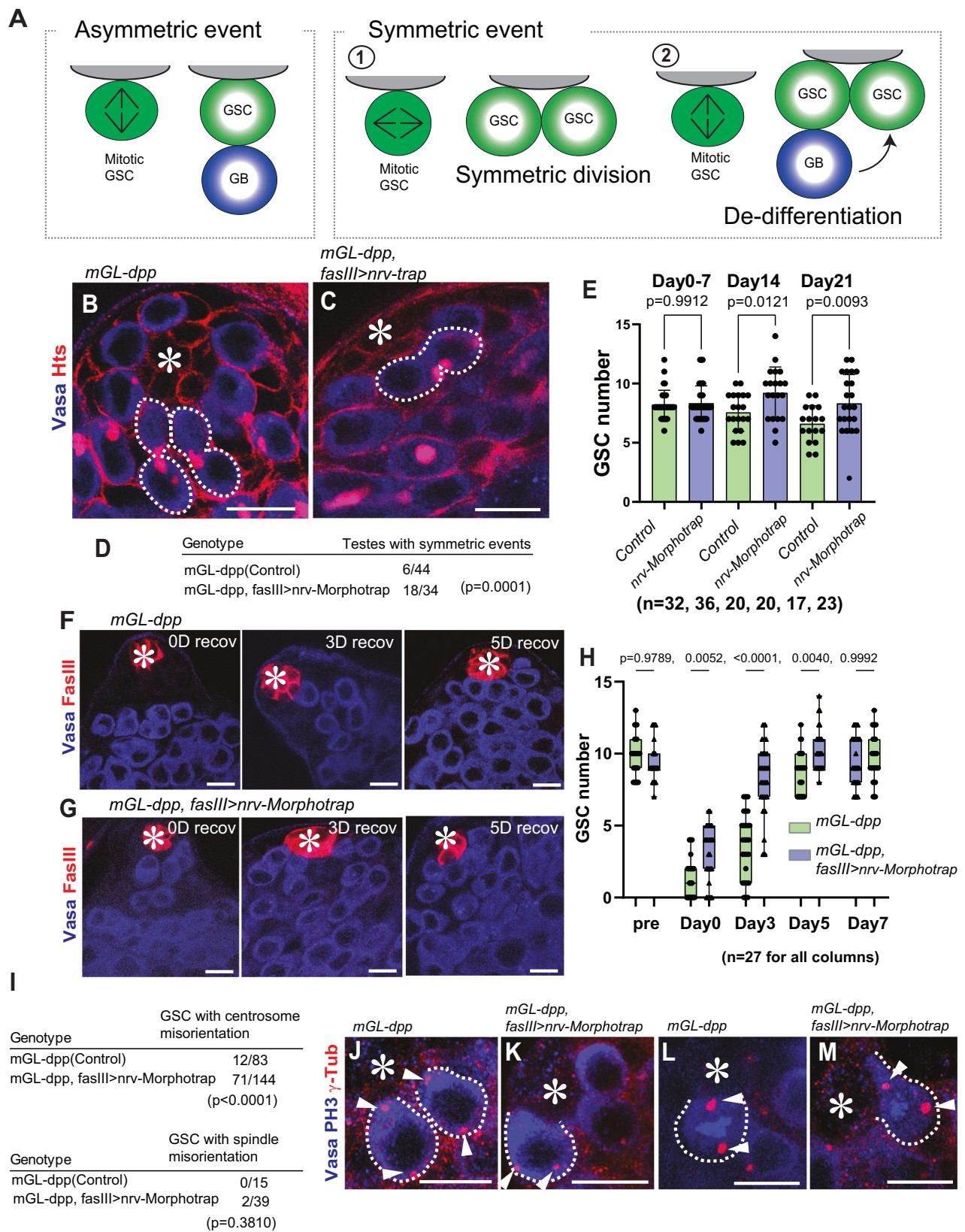

**D**

| Genotype | Testes with symmetric events | |
|---|---|---|
| mGL-dpp(Control) | 6/44 | |
| mGL-dpp, fasIII>nrv-Morphotrap | 18/34 | (p=0.0001) |

**I**

| Genotype | GSC with centrosome misorientation | |
|---|---|---|
| mGL-dpp(Control) | 12/83 | |
| mGL-dpp, fasIII>nrv-Morphotrap | 71/144 | |
| | (p<0.0001) | |

| Genotype | GSC with spindle misorientation | |
|---|---|---|
| mGL-dpp(Control) | 0/15 | |
| mGL-dpp, fasIII>nrv-Morphotrap | 2/39 | |
| | (p=0.3810) | |

knock-in combination – *dppGal4>nrv-Morphotrap, GFP-dpp* CRISPR knock-in homozygous (this line is homozygous viable and no transgenic was used for the rescue) – showed similar effect on the rate of GSC niche replenishing through de-differentiation (Fig. S3), confirming that the trapping of diffusible Dpp is the cause of the observed phenotype.

These data suggests that diffusible Dpp plays a role in preventing de-differentiation in differentiating GBs and SGs. Because Nrv-Morphotrap only affected differentiating cells located away from the niche, but not GSCs within the niche, the data support the idea that increased de-differentiation, rather than increased spindle misorientation (i.e., symmetric division), is responsible for the observed

**Fig. 3 | The diffusible fraction of Dpp prevents de-differentiation. A** Asymmetric and symmetric outcomes of GSC division. Symmetric outcome is defined as the case in which two daughter cells from a GSC division are both placed near the hub, resulting in the production of two GSCs. It occurs as the consequence of either symmetric division[1] or de-differentiation[2] (see details in main text). **B**, **C** Representative images of testis tip without (**B**) or with (**C**) trapping Dpp. Broken lines indicate asymmetric events in (**B**) and a symmetric event in (**C**). **D** Frequency of testes showing any symmetric events (number of testes/scored testes) without or with trapping Dpp. **E** Changes in GSC number across post-eclosion without or with trapping Dpp. **F**, **G** Representative images of testis tip after depletion of GSC by expressing Bam at 0-day recovery (0D recov: immediately after 6-time heat shock treatment) and after 3-day and 5-day recovery time points (3D recov, 5D recov) in room temperature culture without (**F**) or with (**G**) trapping Dpp. **H** Changes in GSC number during recovery from forced differentiation of GSCs

without or with trapping Dpp. Pre: pre-heat shock. **I** Frequency of GSCs (number of GSCs/scored GSCs) with misoriented centrosome or spindle without or with trapping Dpp. **J–M** Representative images of centrosomes in interphase cells (**J**, **K**) and mitotic cells (**L**, **M**) of GSCs without (**J**, **L**) or with (**K**, **M**) trapping Dpp. For trapping Dpp in this figure, Nrv-Morphotrap was expressed under the control of fasIIIGal4 driver in *mGL-dpp* homozygous background rescued by *pPA dpp 8391/X*. Siblings without Gal4 were used for the control. Fixed samples were used for all images and graphs. All scale bars represent 10 μm. Asterisks indicate the approximate location of the hub. n indicates the number of scored testes in **D**, **E** and **H**, or scored GSCs in **I**. Box plots show 25–75% (box), minimum to maximum (whiskers) with all data points. The p-value was calculated by two-tailed student-t-test for **D** and **I** and by Šídák's multiple comparion tests for **E** and **H** and provided on each graph. Source data for **E**, **H** are provided in Source Data file.

high frequency of symmetric events. To further rule out the possibility that defects in spindle orientation in the GSCs contribute to the observation of faster niche replenishment, we assessed the spindle orientations of GSCs in the *mGL-dpp with pPA dpp 8391/X, fasIII>nrv-Morphotrap* flies. We found that GSCs showed correctly oriented spindles (Fig. 3I, M), in support of our conclusions that observed symmetric events are the result of excess de-differentiation and not spindle misorientation. We do observe that although the spindles were correctly oriented, centrosomes of GSCs in *mGL-dpp with pPA dpp 8391/X, fasIII>nrv-Morphotrap* flies were significantly more misoriented (Fig. 3I, M), but this is likely a secondary effect of a higher frequency of de-differentiation as de-differentiated GSCs are reported to have higher instances of centrosome misorientation[31].

### Dpp acts through the same pathway in both cell types

We next asked if Dpp acts through the same signaling pathway in differentiating germ cells as it does in GSCs. Dpp is known to bind to its receptor Thickveins (Tkv) on GSCs and activate signaling to maintain GSC identity[10,11]. Knock-down of Tkv by expression of shRNA under the control of the germline driver nosGal4 results in a depletion of GSCs from the niche (Fig. 4A, B), demonstrating the indispensability of this pathway to GSC maintenance, as consistent with previous reports[10,11].

To determine if Tkv is the receptor for diffusible Dpp outside of the niche, we knocked down Tkv specifically in differentiating germ cells using a bamGal4 driver. Intriguingly, we observed a higher number of GSCs per niche in bam>Tkv RNAi testes as the flies aged post-eclosion (Fig. 4C), similar to what was observed in *mGL-dpp with pPA dpp 8391/X, fasIII>nrv-Morphotrap* (Fig. 3E). Moreover, *bam>tkv RNAi* testes exhibit a higher frequency of symmetric events (Fig. 4D–F), recapitulating the phenotype of *mGL-dpp with pPA dpp 8391/X, fasIII>nrv-Morphotrap* flies. Analysis of *hs-bam with bam>tkv RNAi* showed a significantly faster recovery of GSCs in the niche as compared to control after heat-shock mediated depletion of GSCs (Fig. 4G), indicating that Tkv-mediated signaling in differentiating cells impedes de-differentiation, and suggesting that the Tkv-mediated signaling pathway is also utilized by the diffusible fraction of Dpp in differentiating germ cells.

We next knocked-down Mad, the downstream effector of Tkv-signaling, and Medea, the partner of Mad, using bamGal4-mediated shRNA expression in hs-bam flies. We found that both RNAi conditions showed a significantly faster recovery in GSCs in the niche after heat-shock mediated depletion of GSCs, indicating the Tkv-Mad/Medea mediated canonical Dpp signaling pathway responsible for GSC maintenance is also responsible for preventing de-differentiation in differentiated cells (Fig. 4H).

As was the case with *mGL-dpp with pPA dpp 8391/X, fasIII>nrv-Morphotrap* flies, GSC spindles were not misoriented in *bam>tkv RNAi*, *bam>mad RNAi* or *bam>medea RNAi* genotypes (Fig. 4I), again suggesting that de-differentiation, and not spindle misorientation, is responsible for the increase in symmetric events. Centrosomes for

these genotypes did exhibit misorientation (Fig. 4I) but as noted above, this is a phenomenon frequently seen in GSCs as a consequence of de-differentiation.

To further confirm that the observed faster recovery upon depletion of signal from diffusible Dpp is caused by accelerated de-differentiation, we conducted long-term live imaging of germ cells expressing mCherry-Vasa for 16 hours with or without bam>Tkv RNAi (Fig. S4A–C; Supplementary Movies S1, S2). Strikingly, we found dramatic increase of frequency of de-differentiation events in *bam>tkv RNAi* testes, confirming that the diffusible fraction of Dpp indeed prevents de-differentiation.

These results strongly suggest that both the contact-dependent and independent signals from Dpp (inside and outside of the niche) use same signaling pathway to achieve distinct signaling outcomes.

### Gbb also prevents de-differentiation

It has been reported that another BMP ligand, Glass bottom boat (Gbb), the *Drosophila* BMP7 ortholog, is also required for GSC maintenance in the testis[11]. Therefore, we wondered whether Gbb is also required for preventing de-differentiation. First, to understand expression patterns of Dpp and Gbb, we conducted anti-HA staining of fully functional HA-tagged Dpp and Gbb lines[22,32] to define cell types expressing Dpp and Gbb. In a past report, Dpp mRNA was detected both in hub cells and CySCs[11]. However, we detected HA-tagged endogenous Dpp protein exclusively in hub cells, whereas Gbb is expressed not only in hub cells but also in CySCs and CCs (Fig. 5A–C). mGL-Dpp showed an intracellular pattern of hub localization similar to that of anti-HA staining when extracellular mGL-Dpp signal was removed by fixation (Fig. 5D–F). These data suggest that Dpp and Gbb are expressed in distinct cell types, at least at the protein level.

Next, we trapped HA-tagged Dpp or Gbb to block dispersal using expression of HA-trap, a single-chain variable fragment (scFv) against the HA tag[22], analogous to Morphotrap, under the hub driver fasIII-Gal4. Similar to Morphotrap, HA trap consists of an anti-HA scFv fused to the transmembrane domain of mCD8. Testes expressing the HA-Dpp trap showed significant reduction of pMad in GSCs, similar to what is observed when mGL-Dpp is trapped by mCD8-Morphotrap (Fig. 5G, H, and J). In contrast, knock-down of Gbb in CySCs and CCs under the c587-Gal4[ts] driver did not affect signal in GSCs, although it did affect pMad signal in CCs (Fig. S5A, B). These data suggest that GSCs mainly rely on hub-derived Gbb but not CySC/CC-derived Gbb (Fig. 5I, J, Fig. S5A, B).

Since trapping of the HA-tag on Gbb-HA in hub cells showed a significant reduction of pMad in GSCs, we could not use the HA-trap to test the function of Gbb specifically outside of the niche, just like the case of trapping Dpp by mCD8-Morphotrap (Fig. 2L). Therefore, we tested the involvement of a type I receptor, Saxophone (Sax), which preferentially binds to Gbb[33], and Punt, a common type II receptor for both Dpp and Gbb[34]. Testes where either Sax or Punt is knocked down using the bamGal4 driver exhibited acceleration of de-differentiation,

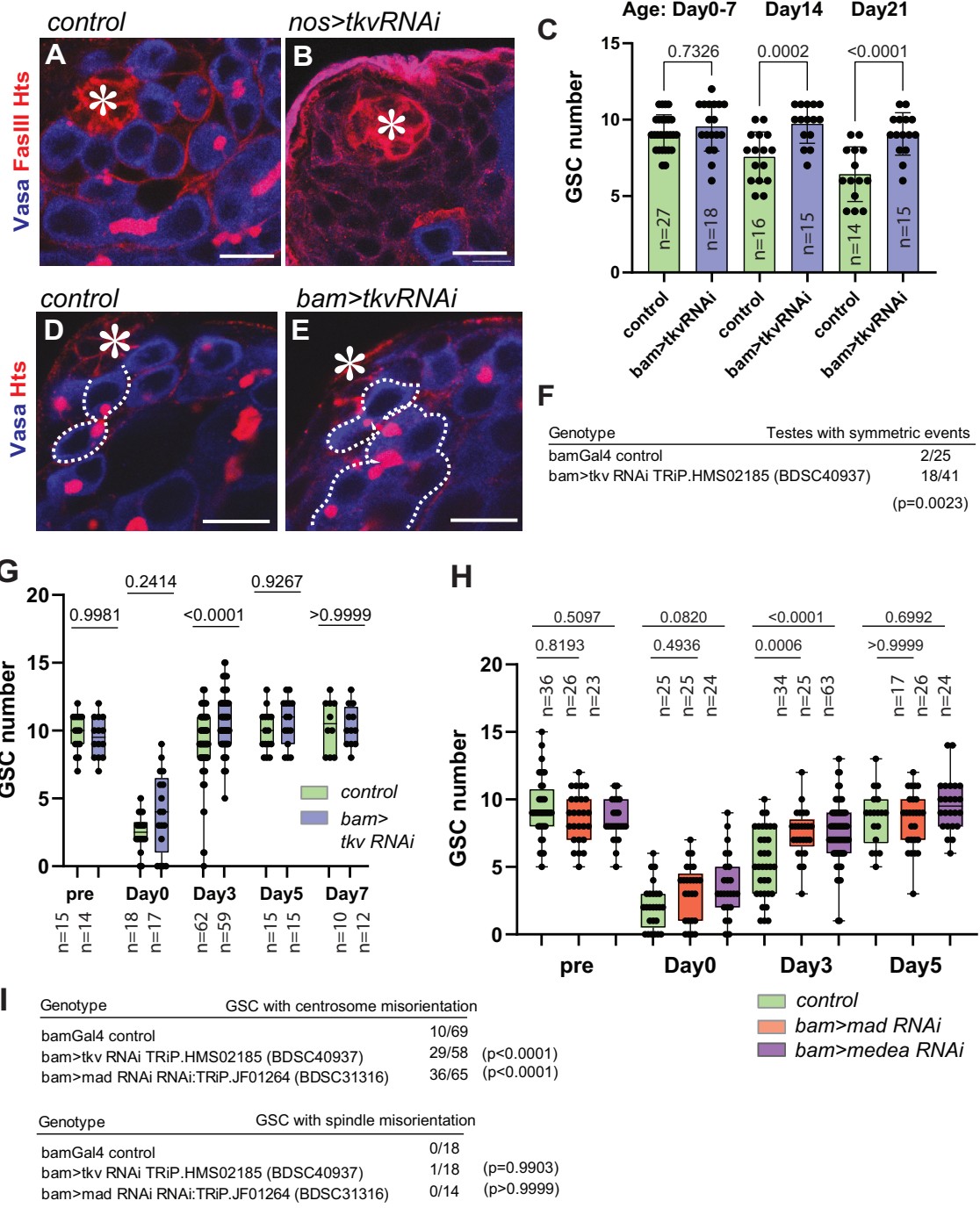

**Fig. 4 | Dpp acts through the same signaling pathway in both cell types.**
**A**, **B** Representative images of testis tip without (**A**) or with (**B**) expression of shRNA against tkv (*tkv RNAi*) under the nosGal4 driver. Tkv RNAi shows the hub without harboring any Vasa positive germ cells. **C** Changes in GSC number during aging without or with *tkv RNAi* expression under the bamGal4 driver. **D**, **E** Representative images of testis tip without (**D**) or with (**E**) *tkv RNAi* under the bamGal4 driver. Broken lines indicate symmetric events. **F** Frequency of testes showing any symmetric events (number of testes/scored testes) without or with *bam>tkv RNAi*. **H** Changes in GSC number during recovery from forced differentiation of GSCs without or with bam>Tkv RNAi (**G**), Mad RNAi, Medea RNAi (**H**). **I** Frequency of GSCs (number of GSCs/scored GSCs) with misoriented centrosomes or spindles in indicated genotypes. All scale bars represent 10 μm. Asterisks indicate approximate location of the hub. Fixed samples were used for all images and graphs. n indicates the number of scored testes in **C**, **F**, **G** and **H**. Box plots show 25–75% (box), minimum to maximum (whiskers) with all data points. The p-values were calculated by two-tailed student-t-tests for **F** and by Šídák's multiple comparisons tests for **C**, **G**, **H**, **I**. Source data for **C**, **G**, **H** are provided in Source Data file.

similar to the phenotypes of mGL-Dpp trap and Tkv RNAi, indicating that both Dpp and Gbb outside of the niche are required for preventing de-diffrentiation (Fig. 5K).

Knock-down of Punt but not Sax under the control of the germline driver nosGal4 resulted in an immediate depletion of GSCs from the niche (Fig. S5C–F), demonstrating that unlike Punt, the requirement of Sax to GSC maintenance is minor. Moreover, pMad was strongly positive in *nos>sax RNAi* GSCs (Fig. S5E, F), suggesting that Sax mainly works outside of the niche.

Based on these results, we propose that both ligands, Dpp and Gbb, may have critical effects on both GSC maintenance and the prevention of de-differentiation. We observed co-localization of Dpp

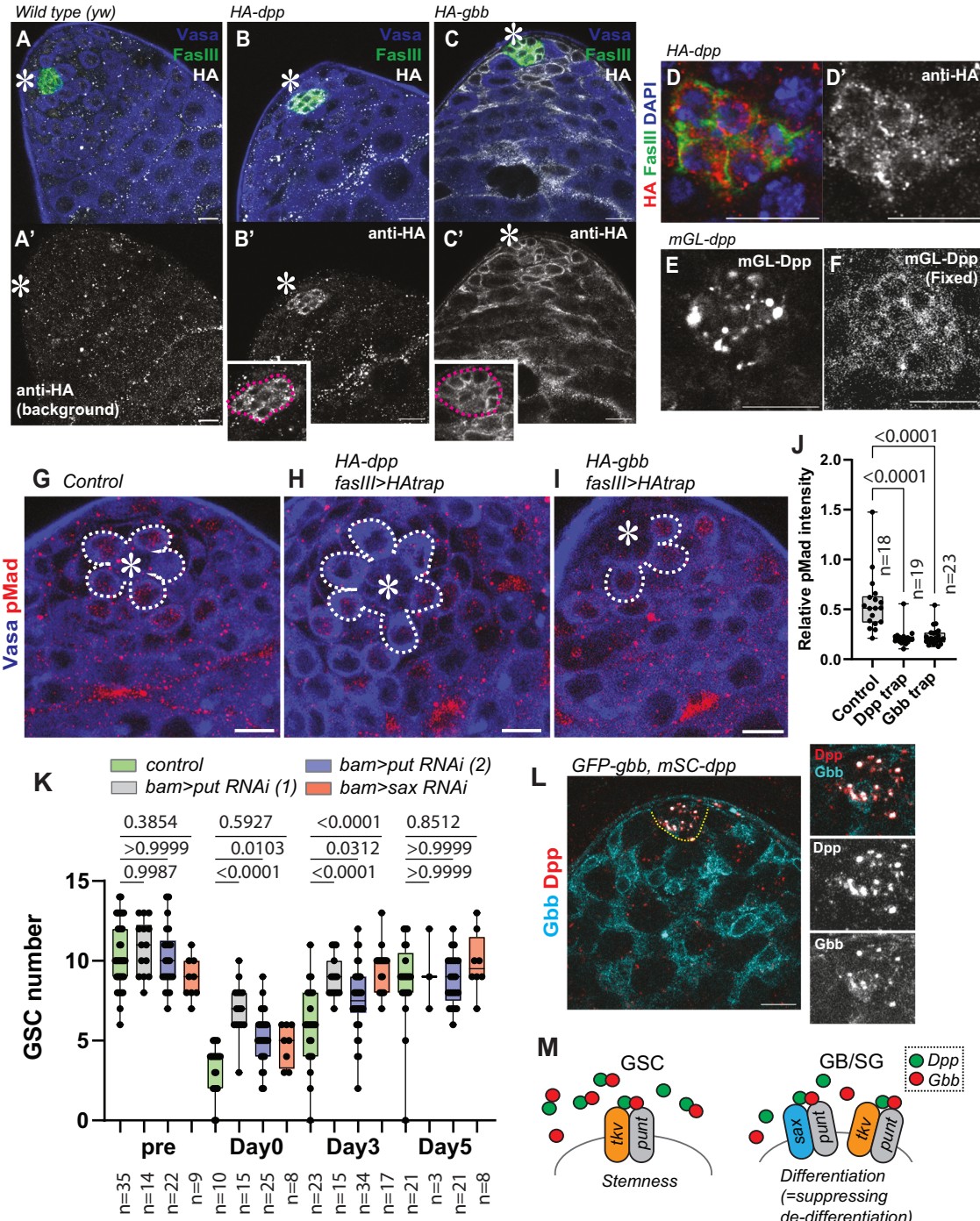

**Fig. 5 | Gbb also prevents de-differentiation. A–C** Representative images of HA staining of indicated genotypes. Homozygous *HA-dpp* and *HA-gbb* flies were used. **A'–C'** shows anti-HA channel. Right panels show magnified areas of the hub. **D** A representative image of the hub of *HA-dpp* testis. **D'** shows anti-HA channel. **E, F** Comparison of mGL-Dpp signal before (**E**) and after (**F**) fixation. **G–I** Representative images of pMad staining of GSCs without (**G**), after trapping Dpp (**H**) or Gbb (**I**) by expressing HAtrap under the FasIIIGal4 driver. GSCs are encircled by white dotted lines. **J** Quantification of pMad intensity in GSCs (relative to CCs) of fasIIIGal4 driven HAtrap in the background of *HA-dpp* homozygous fly or *HA-gbb* homozygous fly. P-values were calculated by Dunnett's multiple comparisons tests. **K** Changes in GSC number during recovery from forced differentiation

of GSCs without or with bam> punt RNAi[1] TRiP.GLV21066, punt RNAi[2] TRiP.HMS01944 or sax RNAi:TRiP.HMJO2118. P-values were calculated by Šídák's multiple comparisons test. **L** A representative image of colocalization of GFP-Gbb and mSC-Dpp in the hub. Images were taken using live testes. **M** Model. Dpp and Gbb may form heterodimer and are secreted from hub cells to induce distinct signaling outcomes in GSC and in GB/SG. Tkv functions both in GSCs and GB/SGs, whereas Sax predominantly functions in GB/SGs. All scale bars represent 10 μm. Asterisks indicate the approximate location of the hub. Box plots show 25–75% (box), minimum to maximum (whiskers) with all data points. n indicates the number of scored GSCs in (**J**), testes in (**K**). Source data for **J, K** are provided in Source Data file.

(mSC-Dpp, TableS1) and Gbb (GFP-Gbb, flytrap BDSC63055) predominantly within the hub, where they may contribute to the formation of sharply graded signaling outcomes in the cells around the niche (Fig. 5L, M). A recent report has demonstrated that heterodimers of Dpp-Gbb form within a cell prior to secretion and are able to trigger strong signaling in the *Drosophila* imaginal wing disc[32]. Similarly, it is possible that the Dpp/Gbb heterodimer may form in the hub and contribute to the graded response of cells in proximity to the hub.

### Diffusible fraction of Dpp enhances *bam* expression

It is known that Dpp signal suppresses expression of the *bam* gene in female GSCs where Bam is necessary and sufficient for differentiation, and its repression in stem cells is essential to maintain their undifferentiated states[35,36]. Although the function of BMP signal on Bam regulation in male GSCs is less clear[13], it appears also to be required for GSC maintenance, at least in part, by repressing *bam* expression[11]. Therefore, we next wondered whether Dpp signal has the opposite effect on *bam* expression in GSCs as compared to differentiating cells.

If the Dpp signal acts on Bam in differentiating cells to inhibit dedifferentiation, Dpp should enhance Bam expression. Thus, we would expect a reduction in Bam expression in differentiating cells if we blocked Dpp diffusion. To test this, we blocked Dpp diffusion using Morphotrap (*fasIII>nrv-Morphotrap* expressed in *mGL-dpp with pPA dpp 8391/X*) and stained these testes for Bam protein. As we expected, Bam expression was reduced in 4-to-8 cell SGs and tended to reach a peak in 16-cell SGs, in contrast to the control in which the peak of Bam expression was seen at 8-cell SGs, as reported previously[37] (representative IF images in Fig. 6A–F). Reduced Bam expression in 4-to-8 cell SGs and the shift of the Bam peak toward 16-cell SGs were also observed when other BMP signaling components were depleted (Fig. 6E, S6). These data suggest that the diffusible fraction of BMP promotes immediate upregulation of Bam in GB-SGs upon exit from GSC stage, opposite of its inhibitory function in GSCs.

Supporting this hypothesis that the Dpp signal has dual roles on *bam* expression, germline tumors induced by expression of the constitutively active form of Tkv (nos>TkvCA) are a mixture of Bam positive and negative cells, as reported previously (Fig. 6G)[13].

To test whether reduced Bam levels in 4-to-8 cell SGs is the major cause of accelerating de-differentiation, we attempted to rescue the bam>Tkv RNAi phenotype by combining it with Bam overexpression. Strikingly, bamGal4-mediated expression of Bam abrogated the observed enhancement of de-differentiation in bam>Tkv RNAi alone (Fig. 6H), indicating that Dpp signal outside of the hub inhibits de-differentiation through immediate upregulation of Bam expression.

### Mad exerts opposing effects on the same target gene *bam*

So far, our data suggest that Dpp/Gbb-Tkv/Punt or Sax/Punt act on Mad/Med to downregulate Bam expression in GSCs but upregulate Bam expression in GB/SGs. How does the same pathway exert opposing effects on the same target gene *bam*?

In the testis, Dpp signal is highest in GSCs, with strong pMad intensity as compared to GB/SGs (Fig. 2I). Therefore, we wondered if the opposing outcomes of Dpp signal depend on pMad concentration. To test the likelihood of this possibility, we took the advantage of nos>TkvCA tumor cells, which express various levels of Bam, to examine relationships between pMad concentration and Bam expression level (Fig. 7A). Interestingly, we indeed found that the pMad levels and Bam levels show a non-linear correlation, such that populations with the highest pMad levels have low levels of Bam, whilst populations with modest pMad levels show the highest levels of Bam (Fig. 7A, B). These two populations mimic GSCs (high pMad, low or no Bam) and SGs (low pMad, high Bam), suggesting that the opposing outcomes of Dpp signaling are likely dependent on pMad concentration.

The Mad binding domain in the *bam* promoter has been well characterized in female GSCs[34,38]. In addition to a previously-characterized Mad binding site required for silencing *bam* in female GSCs (position +39 from the transcription start site, TSS), another putative Mad binding site was reported at position −68 (Fig. 7C)[34,39]. To examine the function of these two sites in male GSCs and differentiating cells, we generated flies carrying bam promoter reporter constructs with or without mutations at these sites that would abrogate Mad binding (Fig. 7C)[34,38]. As reported in female GSCs, we found that the +39 site, but not the −68 site, is required to suppress Bam expression in GSCs, as mutations of +39 site caused precocious expression of the *bam* reporter in GSCs (Fig. 7D, E). Furthermore, we noticed that the +39-site mutant reporter showed drastically lower intensity relative to the control reporter in all stages of SGs (Fig. 7F, G, J), indicating that the +39 site is required for upregulation of *bam* in SGs. In contrast, the −68-site mutant reporter showed increased signal on SGs (Fig. 7F, H, J), indicating that this site has an inhibitory effect on the *bam* promoter in SGs. This effect depends on the +39 site, as the increased signal was not observed when combined with the +39 mutation (Fig. 7I, J). In contrast, activity of the +39 site does not depend on the −68 site, as the reduction of the signal did not change when combined with the −68 mutation (Fig. 7I, J). These data suggest that the +39 site is used by the diffusible fraction of Dpp outside the niche to upregulate *bam* expression (Fig. 7J, K).

It is still unclear how different concentrations of Mad can affect these two binding sites. Mad has been shown to interact with numerous co-factors to act as either a transcriptional repressor or activator[39]. It is possible that Mad interacts stage-specifically with different cofactors on these sites. Moreover, Mad may indirectly regulate these sites through regulating other factors that bind to these sites. Further studies will be required to identify other factors involved in the opposing BMP signal output.

It should be noted that all bam-mGL reporters show uniform expression levels throughout all SG stages (4-to-16 SGs), unlike the endogenous Bam pattern where Bam staining shows peak intensity at 8-cell SG stage. This indicates that downregulation of Bam in 16-SGs is post-transcriptionally regulated. Therefore, we speculate that the observed shift of bam expression peak from 4-to-8 cell SGs to 16-cell SGs in BMP mutants (Fig. 6A-F) is likely caused by a post-transcriptional feedback mechanism in response to reduction of bam transcription in earlier SGs. Further studies will be necessary to fully understand this feedback mechanism.

Taken together, this study provides a new paradigm of niche space restriction. We provide clear evidence that a soluble niche ligand can spread from a niche and facilitates differential signaling responses in stem cells versus their differentiating progenies (Fig. 7L).

## Discussion

In this study, we demonstrate the presence of a diffusible fraction of Dpp and show that it has a key function outside the niche in maintaining GSC daughter cells differentiation, a role opposite to its function in the niche in promoting GSC self-renewal. These opposing signaling outcomes are achieved by the same canonical BMP pathway, i.e. the receptor Tkv, Sax and Punt and the downstream effectors Mad/Medea. The pathway represses Bam expression in stem cells but upregulates Bam expression in differentiating cells. It has been suggested that Dpp has only a minor effect in GSC maintenance in the testicular niche, based on the mild stem-cell loss phenotype of *dpp* mutants[11]. This observation may be explained by opposed function of Dpp in GSC and GB/SGs. Because our findings suggest that Dpp mutants can cause both GSC loss and enhanced de-differentiation at the same time, the balance of these opposing effects could result in an apparently normal GSC number in the niche.

Dpp in the niche has been postulated to act as a highly localized signal, as pMad is observed exclusively in GSCs and their immediate progeny. However, our work has demonstrated the presence of Dpp outside of the niche, implying that the signal reception from niche and

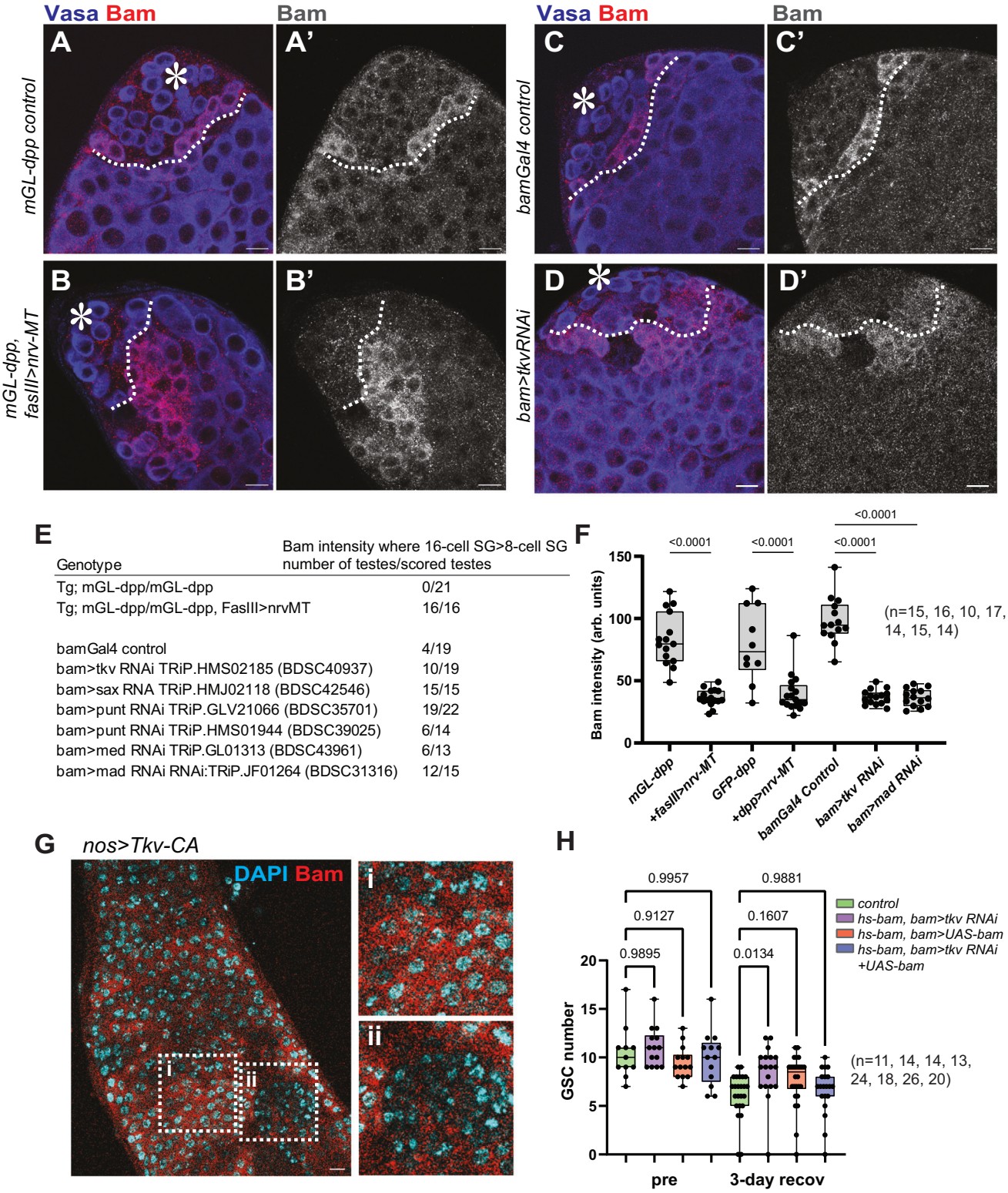

**Fig. 6 | Diffusible fraction of Dpp enhances *bam* expression. A, B** Representative Bam staining images of *mGL-dpp* testis tip without (**A**) or with (**B**) fasIIIGal4 driven Nrv-Morphotrap expression. **C, D** Representative Bam staining images of testis tip without (**C**) or with (**D**) bamGal4 driven tkvRNAi. In **A**–**D**, boundary between 8-cell SGs and 16-cell SGs are divided by white broken lines. **E** Frequency of testes (number of testis/scored testes) in which Bam shows higher expression level in 16-cell SG stages than 8-cell SG stages (see Methods for details of scoring) for indicated genotypes. Tg indicates transgene located on X chromosome (*pPA dpp 8391/X*) for rescuing semi-lethal *mGL-dpp* allele. **F** Quantification of Bam staining intensity in 4-8 cell SGs of indicated reporters. **G** Representative Bam staining images of the tumor cells expressing Tkv-CA under the nosGal4 driver. Right panels are magnification of squared regions of i and ii in the image. **H** Changes in GSC number during recovery after forced niche depletion by GSC differentiation in indicated genotypes. P-values were calculated by Šídák's multiple comparisons test. All scale bars represent 10 μm. Asterisks indicate approximate location of the hub. Fixed samples were used for all images. Box plots show 25–75% (box), minimum to maximum (whiskers) with all data points. n indicates the number of scored testes in (**F**, **H**). Source data are provided in Source Data file. Asterisks indicate approximate location of the hub.

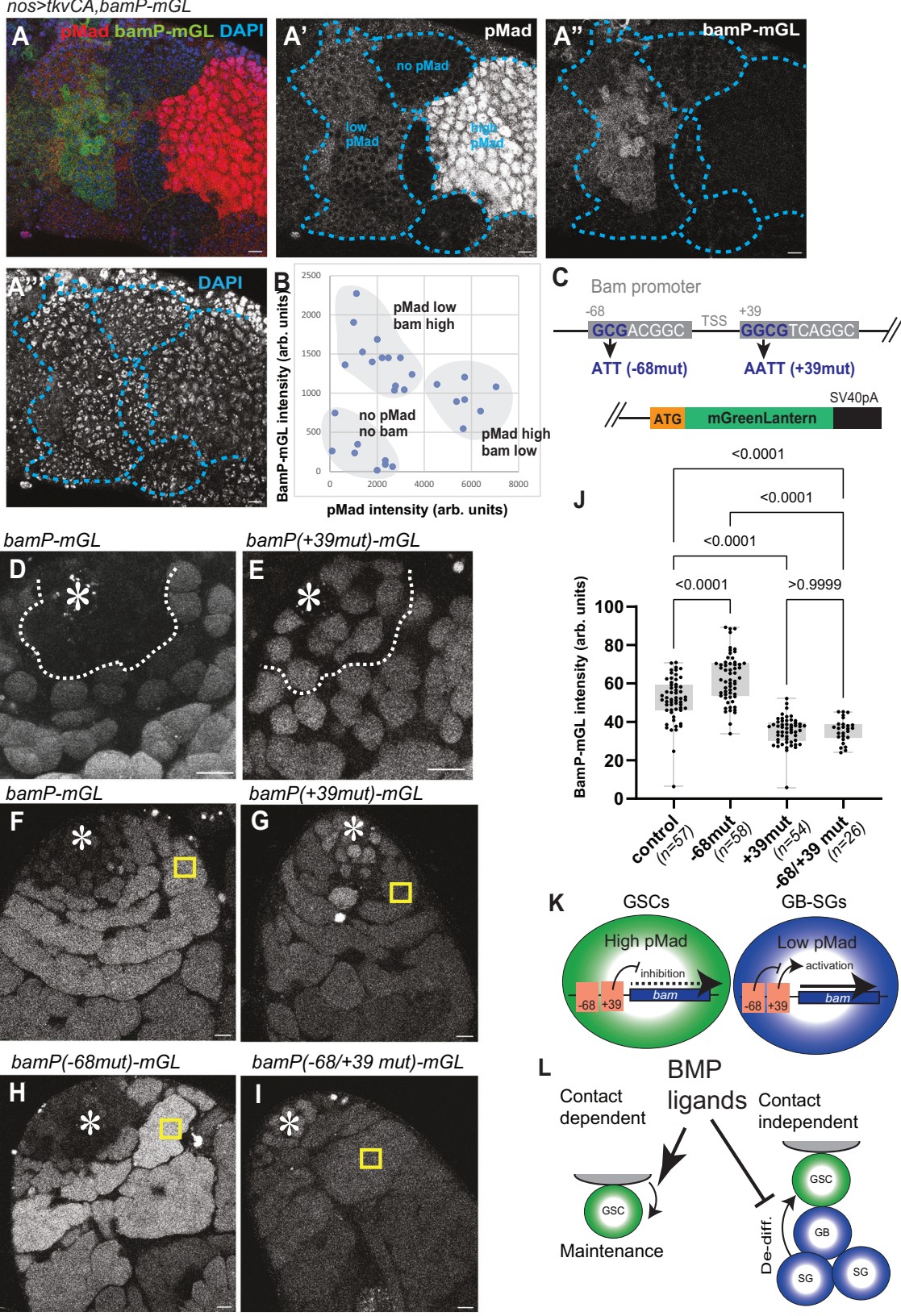

beyond is not uniform. Indeed, many works have revealed how a steep gradient in BMP response is established within just one cell diameter[40–51]. These studies postulated redundant mechanisms in which either stem cells enhance the signal reception, or differentiating cells actively suppress it. Alternatively, it is also possible that the specific composition of ligands are not uniform. Our study suggests a

requirement for both Dpp and Gbb in the observed cellular responses, and we observed co-localization of Dpp-Gbb exclusively in the hub (Fig. 5L). It thus is possible that Dpp-Gbb may form a heterodimer that is more tightly restricted in distribution around the hub, even though each ligand alone can diffuse freely. A recent report has demonstrated that the heterodimers of Dpp-Gbb preferentially form in Dpp

**Fig. 7 | Mad exerts opposing effects on the same target gene *bam*. A** nos>Tkv-CA tumor cells are divided into groups based on pMad intensity by broken lines. **B** Correlation between bam-mGL and pMad intensities. Each measurement was made from a single squared region containing approximately 10-20 tumor cells and background levels were subtracted. **C** Structure of *bam* promoter-mGL reporter constructs. Two putative Mad binding sites are shown in boxes with mutated nucleotides shown in blue. All reporters contain bam promoter from position −198 from TSS to endogenous start codon of bam gene. **D–I** Representative images in the testis of flies harboring indicated bam reporters. **D**, **E** examples of mGL signal in GSCs around the hub in live testes. Asterisks indicate the hub. White broken lines indicate boundary between 2- and 4-cell SGs where Bam staining turns from negative to positive in wild type testis. Wild type reporter, bamP-mGL, shows mGL signal only distal from the line (**D**), whereas mutant reporter, bamP(+39mut)-mGL shows mGL signal in both apical and distal areas (**E**). **F–I** images focus on 4-16 SG stages. Yellow squares are examples of the regions used for quantification of intensities shown in the graph in (**J**). **F** +39 mutant reporter shows lower mGL intensity in SGs. **G** −68 mutant reporter shows higher mGL intensity in SGs. **J** Intensity quantification of indicated reporters in 4-16 cell SGs. P-values were calculated by Šídák's multiple comparisons tests and provided on each graph. Box plots show 25–75% (box), minimum to maximum (whiskers) with all data points. n indicates the number of scored testes. **K** Model. High pMad suppresses bam expression in GSCs, whereas low pMad upregulates bam in SGs through the +39 Mad binding site. −68 site negatively impacts −39 site's effect in SGs. **L** Model. Niche ligands, Dpp and Gbb have effects on GSCs in a contact-dependent manner and on differentiating germ cells (GBs and SGs) through diffusion. The contact-dependent signal is required for stem cell maintenance (Self-renewal), whereas diffusing ligands promote differentiation of daughter cells by preventing de-differentiation (De-diff). Source data for **B**, **J** are provided in Source Data file. All scale bars represent 10 µm. Asterisks indicate approximate location of the hub. Fixed samples were used for **A**, **B**. Live tissues were used for **D–J**.

producing cells and play critical roles during *Drosophila* development[32,52]. This may contribute to the formation of sharply graded signaling outcomes around the niche. It would be interesting to investigate the precise distribution and composition of these ligands in a quantitative manner.

Because mammalian homologs of Dpp, the TGF-beta family genes, broadly regulate processes in many types of stem cell niches[9], we propose that the diffusion of the ligands may be a common mechanism in stem cell niches to ensure their spatial restriction and asymmetric outcome of stem cell division. Intriguingly, differential effects for a BMP ligand, transforming growth factor (TGF)-β, albeit focused on proliferation, have been reported in the mouse hematopoietic stem cell (HSC) niche, where low concentrations of TGF-β induces proliferation of myeloid-biased HSCs but inhibit proliferation of lymphoid-biased HSCs (Ly-HSCs)[53,54]. In this case, it is still unknown whether the ligand forms a gradient around the HSC niche and whether these progenitors are located in distinct positions that subject them to different populations of TGF-β. The elucidation of the basis of these differential outputs based on ligand behavior will be fascinating for future study.

## Methods
### Fly husbandry and strains
Flies were raised on standard Bloomington medium (Lab express) at 25 °C (unless temperature control was required). The following fly stocks were obtained from Bloomington stock center (BDSC); *nosGal4* (BDSC64277); *hs-bam* (BDSC24636); *tkv RNAi* (BDSC40937); Nrv1 Morphotrap (*lexAop-UAS-GrabFP.B.Ext.TagBFP*, BDSC68173); mCD8-Morphotrap (*lexAop-UAS-Morphotrap.ext.mCh*, BDSC68170); *medea RNAi:TRiP.GL01313* (BDSC43961); *mad RNAi:TRiP.JF01264* (BDSC31316); *sax RNAi:TRiP.HMJ02118* (BDSC42546); *punt RNAi: TRiP.HMS01944* (BDSC39025); *punt RNAi: TRiP.GLV21066* (BDSC35701); *gbb RNAi:TRiP.HMS01243* (BDSC34898); *tkv-CA* (BDSC36537); *UAS-GFP.dsRNA.R* (BDSC44415); *gbb-GFP.R* (BDSC63055). *yw* (BDSC189) was used for wildtype. *UAS-GFP-Mad*[51], *HA-dpp*[22], *UAS-HA-trap*[22], *HA-gbb*[32] *dppGal4*[22] lines are described elsewhere. *FasIIIGal4* was obtained from DGRC, Kyoto Stock Center (A04-1-1 DGRC#103-948). *GFP-dpp* and *mCherry-dpp* (FBst0086273) CRISPR knock-in lines were kind gift from Thomas Kornberg and Ryo Hattori[21]. *pVas-Vasa-mCherry* (FBtp0065762)[55], *UAS-histone H3-GFP* and *bamGal4* on 3rd was kind gifts from Yukiko Yamashita.

Temperature shift was performed by culturing flies at room temperature and shifted to 29° C upon eclosion for the 4 days before analysis. Combinations of Tub-Gal80ts (a gift from Cheng-Yu-Lee) with c587Gal4 (a gift from Yukiko M. Yamashita) were used.

For all crosses for obtaining *mGL-dpp* homozygous flies, transgenic allele containing *dpp* locus (*pPA dpp 8391/X*)[23] was introduced to assist embryonic expression and rescue semi-lethality. This transgene has been known to only rescue early development of *dpp* null mutant[23].

### Generation of *mGL-dpp* and *mSC-dpp* alleles
The detail procedure to generate endogenously tagged *dpp* alleles were previously reported[22]. In brief, utilizing the *attP* sites in a MiMIC transposon inserted in the *dpp locus* (*MiMIC dppMI03752*, BDSC36399), about 4.4 kb of the *dpp* genomic sequences containing the second (last) coding exon of *dpp* including a tag and its flanking sequences was inserted in the intron between *dpp*'s two coding exons. The endogenous exon was then removed using FLP-FRT to keep only the tagged exon. mGL (mGreenLantern[56]) or mSC (mScarlet[57]) were inserted in frame after amino acid 485 (NM_164488.2) after the last processing site to tag all the Dpp mature ligands. mGL coding sequences after the last processing site. The detail characterization of these alleles are described in[24].

### Generation of *UASp-bam* transgenic line
*bam* cDNA was PCR-amplified from cDNA pool isolated from wild-type testis (*yw*) using the following primers with restriction sites (underlined):

NotI bam Forward 5′-AC<u>GCGGCCGC</u>ACCATGCTTAATGCACGTG ACGTGTGTC-3′

AscI bam Reverse 5′-AT<u>GGCGCGCC</u>TTAGCTTCTGAAGCGAGGT ACACGTCCGG-3′

PCR products were then digested with NotI and AscI and ligated to a modified pPGW vector (kind gift from Michael Buszczak) using NotI and AscI sites within the multiple cloning site and verified by Sanger sequencing (Genewiz). Transgenic flies were generated using strain attP2 by PhiC31 integrase-mediated transgenesis (BestGene).

### Generation of *bam* reporter transgenic lines
Bam promoter fragment was amplified from genomic DNA using following primers. Overlap sequences for Gibson Assembly reaction were added for each primer.

1) Bam promoter-F: 5′-AGCGGATCCAAGCTTGCATGCGGTACCCC AAATCAGTGTGTATAATT-3′

2) Bam promoter-R: 5′-TATTCTTAAGTTAAATCACACAAATCAC TCGAT −3′ Mad binding site mutation was designed as previously described[36] and introduced by PCR using following primers.

mut-F: 5′-CGCAGACAGCGTAATTTCAGCGATTTCAAACGGTAAA AAG-3′

mut-R: 5′-GAAATTACGCTGTCTGCGAATTCAGGAGAAAGAGGAA GAA-3′

Bam promoter-F/Bam promoter-R fragment was assembled with pUAST-GFP-attB vector (gift from Cheng-Yu-Lee) digested by SphI/NotI to remove UAS promoter located between these sites and mGL fragment.

For Mad-binding site mutant reporter, Bam promoter-F/mut-R, mut-F/Bam promoter-R fragments were assembled with the same vector and mGL fragment.

mGL fragment was amplified from synthesized DNA (below) by using following primers

mGL-forward: 5′-AACTTAAGAATAATGGTGAGCAAGGGCGAGGAGCTGT-3′

mGL-reverse:
5′-TAGAGGTACCCTCGAGCCGCTTACTTGTACAGCTCGTCCATGCCGAGA-3′

mGL gBlock fragment: 5′atggtgagcaagggcgaggagctgttcaccggggtggtgcccatcctggtcgagctggacggcgacgtaaacggccacaagttcagcgtccgcggcgagggcgagggcgatgccaccaacggcaagctgaccctgaagttcatctgcaccaccggcaagctgcccgtgccctggcccacccctcgtgaccaccttaggctacggcgtggcctgcttcgcccgctaccccgaccacatgaagcagcacgacttcttcaagtccgccatgcccgaaggctacgtccaggagcgcaccatctctttcaaggacgacggcacctacaagacccgcgccgaggtgaagttcgagggcgacaccctggtgaaccgcatcgtgctgaagggcatcgacttcaaggagacggcaacatcctggggcacaagctggagtacaacttcaacagccacaaggtctatatcacggccgacaagcagaagaacggcatcaaggctaacttcaagaacccgccacaacgttgaggacggcggcgtgcagctcgccgaccactaccagcagaacacccccatcggcgacggccccgtgctgctgcccgacaaccactacctgagccatcagtccaaactgagcaaagacccccaacgagaagcgcgatcacatggtcctgaaggagagggtgaccgccgccgggattacacatgacatggacgagctgtacaagtaa3′

The amplified fragments were assembled using Gibson Assembly kit (NEB) and verified by Sanger sequencing (Genewiz). Transgenic flies were generated using strain attP40 by PhiC31 integrase-mediated transgenesis (BestGene).

All gBlock fragments and primers used in this study were synthesized by Integrated DNA Technologies (IDT).

### Induction of de-differentiation

Induction of de-differentiation was performed following previously described method with modifications[27]. Approximately 0- to 3-day-old adult flies carrying hs-Bam (BDSC24636) transgene were raised in 22 °C and heat-shocked in a 37 °C water bath for 30 min twice daily in vials with fly food. Vials were placed in a 29 °C incubator between heat-shock treatments. After 6-time treatments, vials were returned to 22 °C for recovery. Testes were dissected at desired recovery time points.

### Short-term live imaging

We used short term live imaging for static image acquisition to observe/quantify fluorescent-tagged proteins to avoid loss of fluorescent signal or tagged protein itself located in extracellular space by fixation and permeabilization.

Testes from newly eclosed flies were dissected into Schneider's Drosophila medium containing 10% fetal bovine serum and glutamine–penicillin–streptomycin. These testes were placed onto Gold Seal Rite-On Micro Slides' 2 etched rings with media, then covered with coverslips. Images were taken using a Zeiss LSM800 airyscan with a 63× oil immersion objective (NA = 1.4), with 10-20 z-stacks (interval 1 μm). For all short-term live imaging experiments, imaging was performed within 30 minutes and no time-lapse imaging was performed using this method.

### Long-term live imaging of de-differentiation

Imaging was performed as previously described[58]. The testes were dissected in 1X Becker Ringer's solution[58] and then mounted onto a 35 mm Glass Bottom Dishes (Nunc). 500 μL of 1 mg/mL poly-L-lysine (Sigma) was pipetted onto the coverslip portion of the imaging dish and incubated for 5- to 7-hours at room temperature. Then, poly-L-lysine solution was replaced to the Becker Ringer's solution and testes were mounted onto poly-L-lysine layer with the tip of the testes oriented toward bottom. Next, Becker Ringer's solution was slowly removed and replaced with 3 ml of room temperature Schneider's Drosophila medium supplied with 10% fetal bovine serum and glutamine–penicillin–streptomycin (Sigma).

Z-stacks (2 μm interval, for 11 stacks) were taken using Zeiss LSM800 airyscan, 1AU-pinhole with 63X oil immersion objective (NA = 1.4) every 10 minutes for overnight (16 hours). Preset tiling function (Zen software, Zeiss) was used for sequential imaging of multiple positions to obtain time-lapse images from 5-to-8 testes per night.

### Immunofluorescence staining

Testes were dissected in phosphate-buffered saline (PBS) and fixed in 4% formaldehyde in PBS for 30–60 minutes. Next, testes were washed in PBST (PBS + 0.2% TritonX-100, Thermo Fisher) for at least 60 minutes, followed by incubation with primary antibody in 3% (or 5% for pMad staining) bovine serum albumin (BSA) in PBST at 4 °C overnight. Samples were washed for 60 minutes (three times for 20 minutes each) in PBST, incubated with secondary antibody in 3% BSA in PBST at room temperature for 2 hours and then washed for 60 minutes (three times for 20 minutes each) in PBST. Samples were then mounted using VECTASHIELD with 4′,6-diamidino-2-phenylindole (DAPI) (Vector Lab). For pMad staining, testes were incubated with 5% BSA in PBST for 30 min at room temperature prior to primary antibody incubation to reduce background.

The primary antibodies used were as follows: rat anti-Vasa (RRID: AB_760351, 1:20; DSHB); mouse anti-Hts (1B1; RRID: AB_528070, 1:20; DSHB); mouse-anti-FasIII (RRID:AB_528238, 1:20, 7G10; DSHB); mouse anti-γ-Tubulin (GTU-88; RRID:AB_532292, 1:400; Sigma-Aldrich); Rabbit anti-pMad (RRID:AB_491015, 1:300; Cell Signaling Technology, Cat# 9516); Mouse anti-phospho-Histone H3 (Ser10), clone 3H10 (RRID:AB_477061; 1:200, Sigma-Aldrich); Rabbit anti-HA C29F4 (RRID:AB_1549585, 1:300, Cell Signaling Technology, Cat# 3724). AlexaFluor-conjugated secondary antibodies (Abcam) were used at a dilution of 1:400.

For Bam staining, 0.2% Tween-20 (Thermo Fisher) was used instead of TritonX-100 for PBS-T. mouse anti-Bam (1:20) antibody was a kind gift from Michael Buszczak.

Images were taken using Zeiss LSM800 confocal microscope with airyscan module by using 1AU-pinhole with 63X oil immersion objective (NA = 1.4). Images were processed by image J/FIJI.

### Chloroquine treatment

Testes from newly eclosed flies were dissected into Schneider's Drosophila medium containing 10% fetal bovine serum and glutamine–penicillin–streptomycin with or without 100 μM chloroquine (Sigma) and incubated for 4 hours at room temperature. These testes were placed onto Gold Seal Rite-On Micro Slides' 2 etched rings with media, then covered with coverslips. An inverted Zeiss LSM800 airyscan with a 63× oil immersion objective (NA = 1.4) was used for imaging.

### Fluorescence recovery after photobleaching (FRAP)

Testes from newly eclosed flies were dissected into Schneider's Drosophila medium containing 10% fetal bovine serum and glutamine–penicillin–streptomycin. These testes were placed onto Gold Seal Rite-On Micro Slides' 2 etched rings with media, then covered with coverslips. Images were taken using a Zeiss LSM800 confocal microscope with a 63× oil immersion objective (NA = 1.4) within 30 minutes. For all live imaging experiments, imaging was performed within 30 minutes. Fluorescence recovery after photo-bleaching (FRAP) of mGL Dpp signal was undertaken using a Zeiss LSM800 with airyscan module by using 1AU-pinhole with 63X oil immersion objective (NA = 1.4). Zen software was used for programming each experiment. Encircled areas of interest (randomly chosen 5μm-diameter circles from the area within less than 40 μm away from the testis tip) were photobleached using the 488 nm laser (laser power; 100%, iterations; 10). Fluorescence recovery was monitored every 10 seconds for single z-plane. Background signal taken in outside of the tissue in each time point were subtracted from the signal of bleached region. Dextran dye permeabilization assay was performed

as described previously[59]. Briefly, testes were incubated with 10 kDa dextran conjugated to AlexaFluor 647 (Thermo Fisher, Catalog number: D22914) at a final concentration of 0.2 μg/μl in 1 mL media for 5 min then immediately subjected for imaging within 30 min. The acquisition setting was adjusted for each sample and normalized by calculating % recovery rate.

Images were processed by image J/FIJI. %recovery rate was calculated as follows; Let $I^t$ be the intensity at each time point (t), $I^{post}$ be the intensity at post-bleaching (first postbleach scan) and $I^{pre}$ be the intensity at pre-bleaching. The governing equation of % recovery is: % recovery= $(I^t - I^{post})/ (I^{pre} - I^{post})$ x 100.

### Quantification of pMad intensities
Image-J/Fiji software was used for image quantification. Average intensity was measured for anti-pMad staining from each GSC nucleus using a single slice from a z-stack series taken by 1AU-pinhole with 63X/1.4 NA oil objective confocal imaging, and background level measured distal region of the same testis was subtracted. The same acquisition setting was used across the samples. To normalize the staining conditions, the average intensities of pMad from four CCs in the same testes were used as internal control and the ratios of intensities were calculated as each GSC per average intensities of CC. The means and s.d. were plotted to the graph for each genotype.

Mean intensity values (a.u.) of CCs were unchanged for genotypes shown in Fig. S2D–G (see details in main text).

### Quantification and staging of Bam expression and Bam reporter intensities
Image-J/Fiji software was used for quantification. Average intensity was measured for anti-Bam staining or mGL signal from regions of 4- or 8-cell SG cysts or 16-cell SG cysts using a single z-stack (1 μm interval) taken by 1AU-pinhole with 63X/1.4 NA oil objective, and subtracted background measured from distal area of the testis within the same slice. Same acquisition setting was used across the samples. For Fig. 6E, average intensities of measurement of 3 portions each from 8-cell SG cysts and 16-cell SG cysts were measured and subtracted the background (taken from distal area of the same testis), then plotted to calculate the ratio of 16-cell SG intensity/8-cell SG intensity for each testis. When the ratio was greater than 1, the testis was judged for Bam intensity=8 < 16.

### Scoring of centrosome and spindle orientation
Centrosome misorientation was indicated when neither of the two centrosomes were closely associated with the hub-GSC interface during interphase. Spindle misorientation was indicated when neither of the two spindle poles were closely associated with the hub-GSC interface during mitosis.

### Statistics and reproducibility
No statistical methods were used to predetermine the sample size. The experiments were not randomized. The investigators were not blinded to allocation during experiments and outcome assessment. All experiments were independently repeated at least 3 times to confirm the results. Statistical analysis and graphing were performed using GraphPad Prism 9 software.

The catalog number of call commercial reagents used in this study is provided in Supplementary Data 1.

### Source data
Individual numerical values displayed in all graphs are provided.

### Reporting summary
Further information on research design is available in the Nature Portfolio Reporting Summary linked to this article.

## Data availability
The data that support all experimental findings of this study are available within the paper and its Supplementary Information files and in the BioStudies database under the accession number S-BIAD1008. Source data are provided with this paper.

## Code availability
Confocal images were collected by a Zeiss LSM 800 Confocal microscope using ZEN software. Confocal images were analyzed using ImageJ/Fiji software (version 2.1.0). Statistical analysis and figures were generated using GraphPad Prism software (version 9.2.0). Illustrator (24.1.3) was used for figure preparation. Raw data necessary to reproduce all statistical analyses and results in the paper are provided in the Source Data file.

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

## Acknowledgements

We thank Markus Affolter, Margaret T. Fuller for discussion and suggestions. Yukiko Yamashita, Michael Buszczak, Thomas Kornberg, Ryo Hattori, and the Bloomington Drosophila Stock Center and the

Developmental Studies Hybridoma Bank for reagents; Marie Bao (Life Science Editors) for manuscript editing. This research is supported by R35GM128678 from the National Institute for General Medical Sciences and a start-up fund from UConn Health (to M.I.) and an SNSF Ambizione grant (PZOOP3_180019) (to S.M.).

## Author contributions

M.I. conceived the project. M.I., S.M.R., A.T., and E.K.B. designed and executed experiments and analyzed data. S.M. generated mGL-dpp line provided all Morphotrap and tagged dpp or gbb lines, assisted with the design of the experiments. S.P. generated bam reporter flies. M.B.B. generated UASp-bam transgenic fly. A.E.C. quantitative analysis and interpretation of imaging data. M.I., M.A., S.M., and S.M.R. drafted the manuscript. All authors edited the manuscript.

## Competing interests

The authors declare no competing interests.
