## [Peer Review File · Nature Communications]

Diffusible fraction of niche BMP ligand safeguards stem-cell differentiationREVIEWER COMMENTS

Reviewer #1 (Remarks to the Author):

Key Results

The authors provide evidence showing a diffusible fraction of Dpp present at greater distances from the niche than was previously known. They do so by using a *Drosophila* line in which the endogenous *dpp* gene is replaced with a GFP-tagged version under control of endogenous regulatory elements. This GFP-tagged Dpp ligand is clearly present in extracellular space between germ cells many cell tiers beyond the row of germline stem cells (GSCs). After identifying this diffusible Dpp, the authors show that they can use a Morphotrap (MT) tool to trap secreted GFP-tagged ligand at the cell membrane, enabling specific genetic manipulation of only diffusible Dpp ligand. The authors provide evidence that diffusible Dpp signals through the Tkv receptor to prevent de-differentiation in germ cells further from the niche.

A role for preventing de-differentiation is striking given what is known about Dpp preventing differentiation in GSCs. To explain these opposing roles of Dpp signaling in germ cells, the authors provide support for concentration dependent activity of pMad at the *bam* promoter region. The authors tested two core pMad binding sequences. Their results support a role for binding of pMad at both sites at high concentrations to inhibit *bam* expression near the niche. At further distances from the niche where pMad concentration is lower, the authors note a role of the +39 binding site in enabling pMad activation of *bam* expression. Their results suggest that high concentrations of pMad suppress differentiation by repressing *bam*; low concentrations of pMad promote differentiation and *bam* expression.

Significance

This work should have high significance to both its field and related fields. The work supports a novel role for an important signaling pathway relevant to tissue homeostasis in the *Drosophila* testis. Dpp signaling is well known for being important to maintain GSCs in an undifferentiated state. The manuscript at hand identifies an opposing role for Dpp in promoting differentiation in germ cells that are not at the niche. In addition, and related to this finding, this manuscript identifies a diffusible fraction of Dpp acting much further from the niche than was previously identified. Together, these results reveal previously unknown roles for Dpp signaling in different tissue regions, refining the model for Dpp promoting tissue homeostasis in the testis.

The *Drosophila* testis is a paradigmatic model for niche-stem cell interactions and has repeatedly elucidated novel concepts in stem cell biology. This concept of a signal that maintains stem cell identity acting to promote differentiation in cells away from the niche should thus prove enlightening across fields related to signaling and tissue homeostasis.

Validity of Data

Conclusions and claims of this manuscript are well supported by the data.

The authors provide careful data in support of a diffusible fraction of Dpp away from the niche that they can manipulate with specificity. The authors (1) show convincing images of Dpp-GFP located in extracellular space between germ cells away from the niche and (2) quantify the fraction of Dpp that is mobile using fluorescence recovery after photobleaching (FRAP) assays. The authors also indicate how they carefully chose a specific Morphotrap tool *nrv*-MT to manipulate diffusible Dpp, which they

show leaves contact-dependent Dpp signaling at the niche unperturbed.

As a suggestion to strengthen the evidence for nrv-MT to tether Dpp-GFP to the niche, it would be helpful to include a higher magnification image of the niche in Figure 2. Figure 2E-F show reduction in the diffusible fraction of Dpp away from the niche while using Morphotrap, but do not show Dpp tethered to the niche. High magnification images in Figure S2 clearly show tethered Dpp. It would help to show images at similar magnification of control and Morphotrap niches to appreciate how the Morphotrap tool increases the concentration of Dpp localized to the niche region.

The data indicating a role for diffusible Dpp in promoting germ cell differentiation is thoroughly supported. When Dpp is tethered to the niche and cannot diffuse, the authors identify a significant increase in de-differentiation events in which gonialblasts revert to GSC fate and occupy positions adjacent to the niche. They observe these increased de-differentiation events when diffusible Dpp is disrupted in aging testes, and in testes in which bam is overexpressed to force GSC differentiation. The authors also show an increase in de-differentiation when the Dpp receptor Tkv is disrupted in germ cells further from the niche. Their results suggest that diffusible Dpp signaling through the Tkv receptor normally preserves differentiation among gonialblasts and other more differentiated germ cells. The authors explain that while de-differentiation has been shown to replenish lost stem cells at the testis niche (Herrera & Bach, 2018), excess de-differentiation can lead to tumorigenesis in other systems. It would be interesting if future work identified the cell biological problems caused by excess de-differentiation in the adult testis, as those findings would further elucidate the protective role played by diffusible Dpp. Those future experiments are beyond the scope of this individual study.

The role of diffusible Dpp in preventing de-differentiation is further supported by additional data examining expression of the differentiation marker Bam. The authors find that blocking Dpp diffusion significantly lowers expression of Bam in germ cells more than 1 cell tier away from the niche. This manuscript presents a specific analysis of pMad binding sites within the bam promoter region, supporting direct binding of pMad at the +39 region as a mechanism for activating bam downstream of Dpp signaling. Together, these results provide strong evidence of a mechanism by which diffusible Dpp signaling could prevent de-differentiation among germ cells distant from the niche.

Methodology & Analysis

Methodology is sound and extremely sophisticated. All data are rigorously quantified using appropriate statistical analyses. All conclusions are supported by convincing statistically significant data. For full transparency, it could be helpful to include exact p-values for any tests in which $0.01 < p < 0.05$ in figure legends.

The methods section is well written, including commendable detail describing how fluorescence intensity was measured. The methods section only requires addition of minor details to allow full reproducibility. Within the "Immunofluorescence Staining" section, the catalog number for the Rabbit anti-pMad antibody from Cell Signaling should be added. The "Live imaging" section could be improved by stating frequency of imaging timepoints and size of Z-stack slices, as this information would help with reproduction of imaging longevity in live fluorescent tissue. A reference to the live imaging protocol that is being followed would also be helpful.

Reviewer #2 (Remarks to the Author):

The article by Ridwan et al demonstrates how the important stem cell maintenance factor Dpp, a BMP ligand, not only diffuses beyond the stem cell niche domain (where for a long time it was thought to be localized) but has antagonistic roles into differentiating spermatogonia. It is very well written, logical and innovative. The authors perform clear and well thought out experiments, and propose a

novel and very interesting mechanism of action for how a single molecule (Dpp) can act in antagonistic manners in stem and differentiating cells, which can be relevant for several other stem cell systems where BMPs are utilized.

From the perspective of this reviewer, there are no major flaws in the current version of the manuscript. There are, however, some points to be considered worth revising for the overall improvement of this manuscript.

Major points:

1-The manuscript is solely focused on the role of Dpp in the testis. PMID: 14973292 however described that Gbb is required, while Dpp is sufficient, for the effects in GSC maintenance. Given the dual role of Dpp in GSCs vs differentiating spermatogonia, it would be ideal if the authors would have investigated – or at least addressed - the potential for the different roles of these two BMP ligands in the different cell types. Similarly, although tkv is the main type I receptor expressed, there are type II receptors that could be contributing differently in stem vs differentiating cells. Given the differential roles of BMP activation in stem vs differentiating cells, I would encourage the authors to at least consider these possibilities further in their manuscript.

2-The authors decide to circumvent the fact that mGL-dpp is homozygous lethal by inserting yet another dpp transgene. Why not using mGL-dpp in trans? If the reason behind it is because the mGL signal would be too weak to visualize, it needs to be stated in the text. If another more concerning reason is present, then also explain.

3-“It has been hypothesized that de-differentiation is required for GSC maintenance as flies age (27).” Though still important to check and follow up, PMID: 29985130 suggests de-differentiation doesn't contribute to GSC maintenance under normal aging conditions. Perhaps rephrase this sentence to acknowledge recent data.

4-The hs-bam experiments are a very good idea. However, one still cannot - beyond reasonable doubt - differentiate between the de-differentiation of spermatogonia or the symmetric division of the remaining niche GSCs, now that they have more space to divide in parallel to the hub. If Fig 3G is truly representative, it looks like you have more GSCs left in the niche when nrv-MT is expressed, which could then promote more GSCs to divide symmetrically instead of favoring de-differentiation. Live imaging of these testes could be a feasible option to investigate this mechanism.

5-Perhaps the most exciting of all experiments are the bam promoter/Mad binding sites assay. Very informative! Physiologically, though, can the authors demonstrate that, upon changing Bam expression levels due to the +39 or -38 mutations, changes in SG transit amplification would occur? Bam not only controls the beginning of transit amplification, but high Bam levels also signal for the end of these divisions (PMID: 20018708). As such, one would expect to encounter more than 16-cell cysts when Bam is downregulated in SG cysts, and perhaps premature expression of spermatocyte markers in SG cysts with less than 16 cells when Bam is upregulated. I understand that the authors have been focusing on the contribution of these cells to de-differentiation. But a more well-established mechanism for Bam is the differentiation of the germline, so looking at the effects in the TA zone would make sense.

Minor points

1-Figure 1D-E, please move * to the side/top so Dpp signal in the hub (likely the strongest and most expected signal from Dpp) can be seen.

2-Figure 1H – why is the mGL-Dpp image here so different in resolution than in E? Are they both reflecting the mGL fluorescence of Dpp? If not, it's not clear what H shows. If so, the discrepancy in resolution between E and H needs to be explained.

3-In all figures with merged channels (S2, 5A-C) – please show separate channels. It's hard to see what's going on only in the merge.

4-Fig 5I in the text = actually Fig 5E

5-“It is known that Dpp signal suppresses expression of the bam gene in GSCs (11)” Contrary to what happens in female GSCs, this is still debatable in males, as Dpp/Mad act in GSCs, not GBs, and Bam protein only shows up in 2-4 spermatogonial cysts.

Reviewer #3 (Remarks to the Author):

In this study, Ridwan et al analyse Dpp distribution during in the *Drosophila* germline stem cell (gSC) niche. In this tissue, which serves as a prime paradigm for stem cell homeostasis, Dpp, produced by somatic cells of the niche has thought to act and maintain GSC that are directly in contact with Dpp-producing cells. Indeed, common knowledge from this system and the related GSC niche of the *Drosophila* ovary is that Dpp has an exclusive short range action and multiple mechanisms ensure that Dpp and Dpp signaling is absent in cells that leave the niche during differentiation. The authors use sophisticated genetic tools to identify a pool of Dpp, which is located far away from the main Dpp producing cells of the niche. Using a number of genetic experiments, they suggest that the low level Dpp is important in the progeny of the GSC, the differentiating spermatogonia cells, and serves to protect these cells for de-differentiating back into GSCs. In addition, by analyzing the promoter of the main differentiation gene (*bam*), they suggest a mechanism by which the gene is repressed at high (in GSCs) and activated at low (in spermatogonia) Dpp concentrations. The topic and the study are both interesting, however, there are several points in the present version that do not add up and clearly weaken the central points of the conclusions drawn:

Major remarks

1. The experiment using the morphotrap (MT) are elegant and indeed point to a pool of Dpp that might be freely available. However, the experiments at hand do not necessarily prove that the source of the SG Dpp are the hub cells as suggested by the authors (see also below). MT expressed at the hub cells could as well eliminate a low-level Dpp signal in the SG areas if this pool is produced by other cells (for example other somatic cells outside of the niche). The motile fraction of the ligand could be effectively and irreversibly captured at the hub cells. The fact that somatic cyst cells are pMad positive is indeed an indication that other sources of Dpp exist. Thus, while Dpp indeed exists outside the niche, the situation might be more simple than suggested by the authors. On top of niche-Dpp that maintains GSC, there is a pool of “systemic” low-level Dpp for additional functions.
2. The authors show (Fig 3) that trapping Dpp on hub cells results in an increase in “symmetric events” and postulate that trapping Dpp away from SGs might increase rates of dedifferentiation. To test this, they set up a sophisticated experiment (used previously to monitor GSC replenishment in the niche): They deplete the niche of GSC by forced, long-lasting expression of the differentiation factor *bam* and then, after shutting down *bam* expression, they measure dedifferentiation by counting the re-establishment of GSCs in the niche over a longer time period. However, when carefully looking at the graph of 3H, it is clear that the MT expressing niche starts with significantly more GSC than the controls (time point “post” which I assume refers to the time point directly after the last pulse of *bam* expression). Thus, this experiment can also be interpreted as follows: Dedifferentiation rates are equal but it takes longer for the controls to catch up as they started with fewer GSC on average. This trend (more GSC at timepoint 0) is also visible in the experiments of Fig 4 (panels G,H) where BMP signal transducers are depleted in SGs. Thus, to me it seems that, for some reason, the niche with trapped Dpp is more resistant to the elimination of GSCs (with the *hs-bam* tool) than the controls. This would refute the model that the low levels of Dpp in SGs repress dedifferentiation.

3. The presence of a Smad binding site in the bam promoter (Fig 6) is not supported experimentally. It is completely unclear which method (I guess some in silico tool) the authors identified the sequence at -68 as a Mad binding site (the relevant citations rather refer to the previously identified site that is essential for bam repression in the GSC (Ref 30, 31)). Similarly, occupation of the identified sequence by Mad (and loss of Mad binding in the mutants) is not presented. Thus, the results of the reporter analyses are, at this point a correlation. In addition (but I have had difficulties to understand the exact methods here and I might have missed something), I find the construction of the reporters both unconventional and awkward. If I got this right, those are not neutral reporters that would allow for quantitative comparison of an inert marker (GFP) in an otherwise wild-type tested. They do all contain a bamGFP fusion, which when expressed (at different levels and patterns due to the introduced mutations) will also skew the relative cellular composition of the tissue. For example, mutations that result in loss of bam repression in GSC will activate bamGFP in these cells and result in their differentiation and loss. This would make the comparison of the different transgenic reporters very difficult. I would like again to emphasize here that I might have missed something and, hopefully, misinterpreted the experiment!

Other comments

1. In my opinion, the schematic of 1C (generation of mGL-dpp) makes no sense. The drawing implies that there is some linear intermediate with the given composition, which I do not think is the case. Judging by the description in the legend (the dedicated methods section is not helpful) and the information of Ref20, the recombination event to remove the "last exon of endogenous dpp" is in trans with the PBac(RB) chromosome. Even if there is an error in my reasoning and I am missing a step, the orientation of the FRTs in the construct would catalyze an inversion and not an excision. Please clarify.
2. mGL-Dpp is central to this study, however, the distribution of the molecule is puzzling to me. In a previous paper by the same authors (Ref 17) a similar tool to monitor endogenous Dpp (mCherry Dpp) was used. As expected, the majority of mCherryDpp signal was in the hub cells that produce the fusion protein. In this study, this is not the case: The hub cells can be hardly identified as source cells for mGL-Dpp (by enriched Dpp levels) as would be expected. All we see is a uniform, low-level signal with no signs of enrichment in or near the source (hub). What is the explanation for this, and especially the discrepancy to the previous study? Also, the model suggested by the authors poses that there are two regions with different Dpp signaling levels in the signal receiving cells: high levels at the GSC region and low at the GB/SG area but this is hardly reflected in the distribution of mGL-Dpp.
3. The expression of the morphotrap (MT) results in elimination of mGL signal outside the niche, however it is odd that enrichment of the signal at the membranes of hub cells expressing the MTs is undetectable. This is in contrast to the strong enrichment of mGL signal on membranes of germline cells (nosGal4). What is the reason for this?
4. In connection to the above: The main driver in this study is the newly generated mGLDpp, which seems to be an excellent tool also for other contexts. Given its central role, it is essential to characterize this allele carefully and provide the data (the statement in the methods section "detailed characterization will be reported elsewhere" is not satisfactory). Also, the authors claim that the construct is "semilethal" and flies homozygous for this chromosome need to be supplemented with a transgenic construct that provides Dpp in the embryo. This is somehow at odds with the statement in the methods (line 385) that mGL-dpp homozygous flies show no obvious phenotypes. Do the authors refer to escapers (as there seems to be a fraction that bypasses embryonic lethality without the help of the transgenic Dpp source)? Along these lines, it is important to show that the additional transgenic dpp source is not active during spermatogenesis as this would obscure the findings. I understand that this might be difficult, as it would require some dpp allele that specifically affects GSC maintenance to demonstrate that it cannot be restored by the transgenic Dpp source. Alternatively, the authors might be able to address this by quantitatively showing that homozygous "escapers" males do have normal spermatogenesis (i.e. the transgenic dpp source does not provide ligands during spermatogenesis).
5. The dextran experiment is confusing: The authors state (starting at line 92) that they subjected

testes from mGL-Dpp flies to Dextran incubation. Thus, I would expect to see two panels of the very same tissue/SG area in two different channels (mGLDpp and 10kD-Dextran647) to demonstrate co-localization. Instead, the authors show different tissues and claim that the pattern is similar (i.e. panel E and F are not the same sample and the same is true for panel H and I). Why not monitoring both mGLDpp and 10kD-Dextran647 in the same sample? In addition, the images are not very convincing. Is it possible to use an extracellular staining protocol to exclusively access extracellular fractions of Dpp?

Minor remarks

In general, the draft contains some unclear/clumsy formulations and syntactical flaws and would benefit from a thorough proofreading. Here some examples.

1. The authors use the term "Dpp ligand" (for example line 78) or "Dpp" synonymously throughout the paper. Please use consistently "Dpp" when referring to the ligand. The term "Dpp ligand" is confusing because it can be interpreted as "the ligand of Dpp" implying that Dpp is the receptor.
2. Line 42 : Please specify that the paragraph describes the male germ line stem cell niche
3. Sentence starting at line 92. Please use "we incubated ... in media with freely ..., which is a of similar size... "
4. Introduce pMad ("Phosphorylated Mad (pMad) ..., a read out of Dpp signal activation") earlier: In line 132 instead of 142.
5. Confusing phrasing in paragraph of line 157: The dedifferentiation event in Fig 3A-2 is not a subclass of symmetric GSC division. Change "division" of line 159 with "event"?
6. Fig. 3H: define "post" in the graph. If it is directly after the last heat shock, then it would be better to define it as "0 Days" in recovery
7. Please explain what we see in Fig 4 A and B. I guess the loss of GCSs in B is monitored by the absence of fusome-containing cells ?
8. Please display single channels for bam levels (Fig 5A-C) as the slight differences are masked by the DAPI channel.

Point-by-point responses are provided below.

Reviewer comments are shown in **blue**, our responses are shown in **black**.

Reviewer #1 (Remarks to the Author):

Key Results

The authors provide evidence showing a diffusible fraction of Dpp present at greater distances from the niche than was previously known. They do so by using a *Drosophila* line in which the endogenous *dpp* gene is replaced with a GFP-tagged version under control of endogenous regulatory elements. This GFP-tagged Dpp ligand is clearly present in extracellular space between germ cells many cell tiers beyond the row of germline stem cells (GSCs). After identifying this diffusible Dpp, the authors show that they can use a Morphotrap (MT) tool to trap secreted GFP-tagged ligand at the cell membrane, enabling specific genetic manipulation of only diffusible Dpp ligand. The authors provide evidence that diffusible Dpp signals through the Tkv receptor to prevent de-differentiation in germ cells further from the niche.

A role for preventing de-differentiation is striking given what is known about Dpp preventing differentiation in GSCs. To explain these opposing roles of Dpp signaling in germ cells, the authors provide support for concentration dependent activity of pMad at the *bam* promoter region. The authors tested two core pMad binding sequences. Their results support a role for binding of pMad at both sites at high concentrations to inhibit *bam* expression near the niche. At further distances from the niche where pMad concentration is lower, the authors note a role of the +39 binding site in enabling pMad activation of *bam* expression. Their results suggest that high concentrations of pMad suppress differentiation by repressing *bam*; low concentrations of pMad promote differentiation and *bam*

expression.

Significance

This work should have high significance to both its field and related fields. The work supports a novel role for an important signaling pathway relevant to tissue homeostasis in the *Drosophila* testis. Dpp signaling is well known for being important to maintain GSCs in an undifferentiated state. The manuscript at hand identifies an opposing role for Dpp in promoting differentiation in germ cells that are not at the niche. In addition, and related to this finding, this manuscript identifies a diffusible fraction of Dpp acting much further from the niche than was previously identified. Together, these results reveal previously unknown roles for Dpp signaling in different tissue regions, refining the model for Dpp promoting tissue homeostasis in the testis.

The *Drosophila* testis is a paradigmatic model for niche-stem cell interactions and has repeatedly elucidated novel concepts in stem cell biology. This concept of a signal that maintains stem cell identity acting to promote differentiation in cells away from the niche should thus prove enlightening across fields related to signaling and tissue homeostasis.

We thank for this reviewer for comprehending the significance of this study and providing encouraging comments and suggestions.

Validity of Data

Conclusions and claims of this manuscript are well supported by the data.

The authors provide careful data in support of a diffusible fraction of Dpp away from the niche that they can manipulate with specificity. The authors (1) show convincing images of

Dpp-GFP located in extracellular space between germ cells away from the niche and (2) quantify the fraction of Dpp that is mobile using fluorescence recovery after photobleaching (FRAP) assays. The authors also indicate how they carefully chose a specific Morphotrap tool nrv-MT to manipulate diffusible Dpp, which they show leaves contact-dependent Dpp signaling at the niche unperturbed.

As a suggestion to strengthen the evidence for nrv-MT to tether Dpp-GFP to the niche, it would be helpful to include a higher magnification image of the niche in Figure 2. Figure 2E-F show reduction in the diffusible fraction of Dpp away from the niche while using Morphotrap, but do not show Dpp tethered to the niche. High magnification images in Figure S2 clearly show tethered Dpp. It would help to show images at similar magnification of control and Morphotrap niches to appreciate how the Morphotrap tool increases the concentration of Dpp localized to the niche region.

We appreciate for this important suggestion in evaluating tethered fraction of Dpp in the niche. Similar comment was made by all of the reviewers. We too noted that trapping mGL-Dpp does not seem to give rise to very high mGL-Dpp signal accumulating the surface of hub cells (**new Figure 2H**) as compared with the control (homozygous mGL-Dpp fly without morphotrap, **new Figure 2G**). We therefore wondered if trapped mGL-Dpp may be generally internalized and degraded in hub cells. If so, we should see increased level of Dpp in hub lysosomes when lysosome digestion is perturbed. We found that indeed, the mGL-Dpp signal increased in hub lysosomes after 4-hour chloroquine treatment in the Nrv-MT or mCD8-MT expressing testes compared with control (**new Figure S2**), indicating that trapped Dpp may be constantly degraded in hub cells.

In the previous version of the manuscript, we concluded that mCD8-MT is more actively internalized and degraded, leading to a lower level of pMad in GSCs. However, based on our new data, internalization may be occurring in both morphotrap constructs. Nevertheless,

membrane trapped Dpp with nrv-MT, but not mCD8-MT, appeared to be sufficient to activate signal in GSCs, evidenced by the unchanged pMad level (**new Figure 2I-L, originally 2G-J**). Therefore, we still consider that this condition to be suitable to specifically perturb signal from the diffusing fraction of Dpp outside of the niche while keeping signal intact inside of the niche.

We have now revised the relevant portion as follows:

although we were expecting to see strong signal on the surface of hub cells, the signal was not as high as reflecting equivalent amount to all diffusing fraction of mGL-Dpp. Therefore, we wondered if trapped mGL-Dpp may be constantly internalized and degraded in hub cells. If so, we should see increased level of mGL signal in hub lysosomes when lysosome digestion is perturbed by chloroquine (CQ) treatment, a drug that inhibits lysosomal enzymes. Indeed, the Dpp-mGL signal in hub lysosomes after 4-hour chloroquine treatment showed increase in Dpp-mGL in hub cells of MT expressing testes (Figure S2A-C), indicating that trapped Dpp may be constantly degraded in hub cells.

The data indicating a role for diffusible Dpp in promoting germ cell differentiation is at thoroughly supported. When Dpp is tethered to the niche and cannot diffuse, the authors identify a significant increase in de-differentiation events in which gonialblasts revert to GSC fate and occupy positions adjacent to the niche. They observe these increased de-differentiation events when diffusible Dpp is disrupted in aging testes, and in testes in which bam is overexpressed to force GSC differentiation. The authors also show an increase in de-differentiation when the Dpp receptor Tkv is disrupted in germ cells further from the niche. Their results suggest that diffusible Dpp signaling through the Tkv receptor normally preserves differentiation among gonialblasts and other more differentiated germ cells. The authors explain that while de-differentiation has been shown to replenish lost stem cells at the testis niche (Herrera & Bach, 2018), excess de-differentiation can lead to tumorigenesis in other systems. It would be interesting if future work identified the cell biological

problems caused by excess de-differentiation in the adult testis, as those findings would further elucidate the protective role played by diffusible Dpp. Those future experiments are beyond the scope of this individual study.

Thank you for highlighting this open question, which we agree with the reviewer is of strong interest but beyond the scope of the current study. We understand that the outcome of excess dedifferentiation is still unknown at least in fly testis, and have made this more clear in the revised manuscript. We are highly interested in outcome of excess dedifferentiation and currently working on its effect on clonal expansion of stem cells. We are hoping that we can provide how dedifferentiation affects tissue homeostasis in near future.

The role of diffusible Dpp in preventing de-differentiation is further supported by additional data examining expression of the differentiation marker Bam. The authors find that blocking Dpp diffusion significantly lowers expression of Bam in germ cells more than 1 cell tier away from the niche. This manuscript presents a specific analysis of pMad binding sites within the bam promoter region, supporting direct binding of pMad at the +39 region as a mechanism for activating bam downstream of Dpp signaling. Together, these results provide strong evidence of a mechanism by which diffusible Dpp signaling could prevent de-differentiation among germ cells distant from the niche.

Methodology & Analysis

Methodology is sound and extremely sophisticated. All data are rigorously quantified using appropriate statistical analyses. All conclusions are supported by convincing statistically significant data. For full transparency, it could be helpful to include exact p-values for any tests in which $0.01 < p < 0.05$ in figure legends.

Thank you for pointing this out. Per journal policy, we have now updated all p-values with exact numerical values.

The methods section is well written, including commendable detail describing how fluorescence intensity was measured. The methods section only requires addition of minor details to allow full reproducibility. Within the "Immunofluorescence Staining" section, the catalog number for the Rabbit anti-pMad antibody from Cell Signaling should be added. The "Live imaging" section could be improved by stating frequency of imaging timepoints and size of Z-stack slices, as this information would help with reproduction of imaging longevity in live fluorescent tissue. A reference to the live imaging protocol that is being followed would also be helpful.

We now added this information to the method section.

Reviewer #2 (Remarks to the Author):

The article by Ridwan et al demonstrates how the important stem cell maintenance factor Dpp, a BMP ligand, not only diffuses beyond the stem cell niche domain (where for a long time it was thought to be localized) but has antagonistic roles into differentiating spermatogonia. It is very well written, logical and innovative. The authors perform clear and well thought out experiments, and propose a novel and very interesting mechanism of action for how a single molecule (Dpp) can act in antagonistic manners in stem and differentiating cells, which can be relevant for several other stem cell systems where BMPs are utilized.

From the perspective of this reviewer, there are no major flaws in the current version of the manuscript. There are, however, some points to be consider worth revising for the overall improvement of this manuscript.

We thank this reviewer for his/her encouraging and important comments. Especially, we appreciate this reviewer's guidance on examining the effect of other components of BMP signal, which helped us to realize several important points to add (see below). We have now addressed all comments of this reviewer by performing additional experiments and reanalyzing some of the data.

Major points:

1-The manuscript is solely focused on the role of Dpp in the testis. PMID: 14973292 however described that Gbb is required, while Dpp is sufficient, for the effects in GSC maintenance. Given the dual role of Dpp in GSCs vs differentiating spermatogonia, it would be ideal if the authors would have investigated – or at least addressed - the potential for the different roles of these two BMP ligands in the different cell types. Similarly, although tkv is the main type I receptor expressed, there are type II receptors that could be contributing differently in stem vs differentiating cells. Given the differential roles of BMP activation in stem vs differentiating cells, I would encourage the authors to at least consider these possibilities further in their manuscript.

We appreciate for this great suggestion. We have now analyzed the effects of several other BMP signaling components, including Gbb, Sax and Punt. Essentially, we have found that all of these tested genes have similar effect as the Dpp/Tkv/Mad axis shown in our original analysis (**new Figure 5**).

First to understand expression pattern of Dpp and Gbb, we conducted staining of HA-tagged Dpp and HA-tagged Gpp (both alleles are fully functional). We found that Dpp is almost exclusively expressed in hub cells, whereas Gbb is expressed not only in hub cells, but also in somatic cyst (stem) cells (**new Figure 5A-F**). Interestingly, trapping of Gbb-HA using HAtrap specifically expressed in hub cells was sufficient to reduce pMad in GSCs, whereas knockdown of gbb in other somatic cells under the c587Gal4 driver did not reduce

pMad in GSCs (but it affects pMad in CCs), indicating that hub-derived Gbb but not CySC derived Gbb can signal to GSCs (**new Figure 5G-J, new Figure S5A, B**).

Since trapping of HA-tag on Gbb-HA in hub cells showed significant reduction of pMad in GSCs, we could not use this method to test the specific effect of Gbb on non-stem cells. Therefore, we tested involvement of a type I receptor, Saxophone (Sax), which preferentially binds to Gbb. Knock-down of Sax under bamGal4 driver showed strong enhancement of dedifferentiation (**new Figure 5K**), similar to the phenotype of mGL-Dpp trap and Tkv RNAi, indicating that both Dpp and Gbb outside of the niche are required for preventing dedifferentiation.

In addition, we have knocked down expression of the type II receptor Punt using the bamGal4 driver and observed the same phenotype of increased dedifferentiation (**new Figure 5K**).

Based on these results, we propose that both ligands, Dpp and Gbb have critical effects both on GSC maintenance and prevention of dedifferentiation.

We added these points in revised manuscript which reads as follows:

Based on these results, we propose that both ligands, Dpp and Gbb, may have critical effect both on GSC maintenance and prevention of dedifferentiation. Despite Dpp shows uniform distribution over the entire tissue and Gbb has broad expression pattern, we observed co-localization of Dpp (mScarlet-Dpp) and Gbb (GFP-Gbb) predominantly within the hub, where they may contribute to the formation of sharply graded signaling outcomes in the cells around the niche (Figure 5L, M). A recent report has demonstrated that the heterodimers of Dpp-Gbb form within a cell prior to secretion and that it triggers strong signaling [1]. It is possible that Dpp/Gbb heterodimer may form specifically in the hub and contribute to graded response of cells.

2-The authors decide to circumvent the fact that *mGL-dpp* is homozygous lethal by inserting yet another *dpp* transgene. Why not using *mGL-dpp* in trans? If the reason behind it is because the *mGL* signal would be too weak to visualize, it needs to be stated in the text. If another more concerning reason is present, then also explain.

We realized that our rationale for using each genotype was not sufficiently explained in the manuscript, as a similar concern was also raised by Reviewer 3. We used the homozygous *mGL-dpp* to visualize Dpp diffusion because the same genotype was used for subsequent morphotrap experiments and the signal in heterozygous *mGL-dpp* was too weak to detect diffusing fraction. Because of semi-lethality of homozygous *mGL-dpp* allele, we introduced a transgene *pPA dpp 8391/X*, which contains the genomic region of *dpp* critical for the early embryogenesis [2] but does not rescue the wing phenotypes of *dpp* mutants [3]. We have now provided this explanation in the revised text.

Because *mGL-Dpp* signal was not distinguishable from background fluorescence in the heterozygous flies, we attempted to obtain homozygous flies. However, we found that the homozygous *mGL-dpp* allele was semi-lethal, hence we introduced a transgenic allele (*pPA dpp 8391/X*) containing genomic region of *dpp* expressed only in embryonic stage [2] to assist embryonic development. These rescued *mGL-dpp* homozygous flies were fully viable and able to reach adulthood with no phenotypes observed. These flies allowed us to successfully visualize endogenous Dpp expression and localization in the testis.

We do note that it is important to show that *pPA dpp 8391/X* is not working in the testis to modify *mGL-dpp* trap phenotype. We expressed GFP-IR under the hub driver (*FasIII-Gal4*) in the *pPA dpp 8391/X; mGL-dpp/mGL-dpp* background to specifically knock down *mGL-dpp* (but not *dpp* on *pPA dpp 8391/X* transgene) and found that pMad signal is significantly reduced in GSCs in the niche, indicating that *mGL-Dpp* is functional and *pPA dpp 8391/X*

transgene does not contribute dpp expression in the hub. We now explained this point with **new Figure S2H**, which reads as below.

It should be noted that knock-down of mGL-tagged Dpp in the hub (*pPA dpp 8391/X, fasIII>GFP-IR, mGL-dpp/mGL-dpp*) significantly reduced pMad signal in GSCs, confirming that rescuing transgene (*pPA dpp 8391/X*) does not contribute to the expression in the hub (Figure S2H).

3-"It has been hypothesized that de-differentiation is required for GSC maintenance as flies age (27)." Though still important to check and follow up, PMID: 29985130 suggests de-differentiation doesn't contribute to GSC maintenance under normal aging conditions. Perhaps rephrase this sentence to acknowledge recent data.

We have revised this sentence accordingly.

Although a recent study suggested that the de-differentiation from the Bam-positive lineage (4-16-cell SGs) is required for maintenance of stem cell number only in challenging conditions [4], it was observed constantly from the earlier lineage (GB to 2-cell SGs) even in physiological conditions [5], indicating that de-differentiation is a critical mechanism to maintain stem cell numbers.

4-The *hs-bam* experiments are a very good idea. However, one still cannot - beyond reasonable doubt - differentiate between the de-differentiation of spermatogonia or the symmetric division of the remaining niche GSCs, now that they have more space to divide in parallel to the hub. If Fig 3G is truly representative, it looks like you have more GSCs left in the niche when *nrv-MT* is expressed, which could then promote more GSCs to divide symmetrically instead of favoring de-differentiation. Live imaging of these testes could be a feasible option to investigate this mechanism.

This is a great suggestion. We have now conducted live imaging using bam>Tkv RNAi flies. Strikingly and consistent with our other data, we observe dramatic increase in frequency of dedifferentiation events as compared to control, and not an increase in symmetric division in GSCs. We provide the data in **new Figure S4** and **new supplementary video S1, 2**.

Relevant portion of the text reads as follows:

To further confirm that the observed faster recovery is surely caused by accelerated de-differentiation, we conducted long-term live imaging of bam>Tkv RNAi flies to measure the frequency of de-differentiation events (Figure S4A-C, supplemental video S1, 2). We monitored the division of GSCs expressing mCherry-Vasa for 16 hours. Strikingly, we found dramatic increase of frequency of de-differentiation events in Tkv RNAi expressing testes, indicating that diffusible fraction of Dpp prevents de-differentiation.

5-Perhaps the most exciting of all experiments are the bam promoter/Mad binding sites assay. Very informative! Physiologically, though, can the authors demonstrate that, upon changing Bam expression levels due to the +39 or -38 mutations, changes in SG transit amplification would occur? Bam not only controls the beginning of transit amplification, but high Bam levels also signal for the end of these divisions (PMID: 20018708). As such, one would expect to encounter more than 16-cell cysts when Bam is downregulated in SG cysts, and perhaps premature expression of spermatocyte markers in SG cysts with less than 16 cells when Bam is upregulated. I understand that the authors have been focusing on the contribution of these cells to de-differentiation. But a more well-established mechanism for Bam is the differentiation of the germline, so looking at the effects in the TA zone would make sense.

Thank you for criticism for this important point. We realized that we neglected to carefully examine the bam expression pattern in the previous version of our manuscript. After revisiting our Bam staining images, we realized that preventing BMP signal in SGs causes delay in the Bam expression peak instead of simply reducing overall Bam levels, such that

BMP signal mutants have Bam expression peak in 16-cell SGs instead of in 4-8 cell SGs (shown in **new Figure 6A-E**, and **new Figure S6**). We essentially did not observe any extra division of SGs possibly because mutants can catch up Bam expression in 16-cell SGs. We consider that diffusible BMP has a role in regulating the timely initiation of bam transcription. We have corrected description of Bam expression pattern in the text, which reads as follows:

If the Dpp signal acts on Bam in differentiating cells to inhibit de-differentiation, Dpp needs to enhance Bam expression. Thus, blocking Dpp diffusion would be expected to show delay or reduction in Bam expression in differentiating cells. To test this, we blocked Dpp diffusion using Morphotrap (*fasIII>nrvMT* in *mGL-dpp* homozygous background, rescued by *pPA dpp 8391/X*) and stained these testes for Bam protein. As we expected, Bam expression was delayed and tended to reach a peak in 16-cell SGs, in contrast to the control in which the peak of Bam expression was seen at 8-cell SGs (Figure 6A-E) as reported previously [6], suggesting that Mad promotes timely upregulation of Bam in GB-SGs upon exit from GSC stage, opposite of its inhibitory function in GSCs (Figure 6A-D, arrowheads and F, and Figure S6A-J, arrowheads).

Minor points

1-Figure 1D-E, please move * to the side/top so Dpp signal in the hub (likely the strongest and most expected signal from Dpp) can be seen.

We now provide higher magnification images for hub area showing trapped mGL-Dpp (**new Figure 2D-H, originally in Figure 2D-F and S2**). Puzzlingly, we do not see strong signal from trapped mGL-Dpp on hub cells but through additional experiments, we have found that mGL-Dpp might be constantly internalized and degraded in hub cells We have added new data and description for this point (See response to Reviewer1, point 1).

2-Figure 1H – why is the mGL-Dpp image here so different in resolution than in E? Are they

both reflecting the mGL fluorescence of Dpp? If not, it's not clear what H shows. If so, the discrepancy in resolution between E and H needs to be explained.

We apologize that the original figures were taken in different magnification/resolutions. We have updated the images so that they are now comparable between different magnifications.

3-In all figures with merged channels (S2, 5A-C) – please show separate channels. It's hard to see what's going on only in the merge.

We have now updated **Figures 2G-H (originally in Figure S2)** and **6A-D, S6 (originally 5A-C)**, showing separate channels.

4-Fig 5I in the text = actually Fig 5E

Corrected, thank you!

5-"It is known that Dpp signal suppresses expression of the bam gene in GSCs (11)"
Contrary to what happens in female GSCs, this is still debatable in males, as Dpp/Mad act in GSCs, not GBs, and Bam protein only shows up in 2-4 spermatogonial cysts.

We have updated the description, which now reads as follows:

It is known that Dpp signal suppresses expression of the bam gene in female GSCs where Bam is necessary and sufficient for differentiation and its suppression in stem cells is essential to maintain their undifferentiated states in female GSCs [7, 8]. Although the function of BMP signal on Bam regulation in male GSCs is less clear [9], it also appears to be required for GSC maintenance at least in part, by repressing bam expression [10]. Therefore, we next wondered

whether Dpp signal has the opposite effect on Bam expression in GSCs as compared to differentiating cells.

Reviewer #3 (Remarks to the Author):

In this study, Ridwan et al analyse Dpp distribution during in the Drosophila germline stem cell (gSC) niche. In this tissue, which serves as a prime paradigm for stem cell homeostasis, Dpp, produced by somatic cells of the niche has thought to act and maintain GSC that are directly in contact with Dpp-producing cells. Indeed, common knowledge from this system and the related GSC niche of the Drosophila ovary is that Dpp has an exclusive short range action and multiple mechanisms ensure that Dpp and Dpp signaling is absent in cells that leave the niche during differentiation. The authors use sophisticated genetic tools to identify a pool of Dpp, which is located far away from the main Dpp producing cells of the niche. Using a number of genetic experiments, they suggest that the low level Dpp is important in the progeny of the GSC, the differentiating spermatogonia cells, and serves to protect these cells for de-differentiating back into GSCs. In addition, by analyzing the promoter of the main differentiation gene (bam), they suggest a mechanism by which the gene is repressed at high (in GSCs) and activated at low (in spermatogonia) Dpp concentrations. The topic and the study are both interesting, however, there are several points in the present version that do not add up and clearly weaken the central points of the conclusions drawn:

We would like to thank this reviewer for his/her encouraging and very insightful comments and great suggestions.

Major remarks

1. The experiment using the morphotrap (MT) are elegant and indeed point to a pool of

Dpp that might be freely available. However, the experiments at hand do not necessarily prove that the source of the SG Dpp are the hub cells as suggested by the authors (see also below). MT expressed at the hub cells could as well eliminate a low-level Dpp signal in the SG areas if this pool is produced by other cells (for example other somatic cells outside of the niche). The motile fraction of the ligand could be effectively and irreversibly captured at the hub cells. The fact that somatic cyst cells are pMad positive is indeed an indication that other sources of Dpp exist. Thus, while Dpp indeed exists outside the niche, the situation might be more simple than suggested by the authors. On top of niche-Dpp that maintains GSC, there is a pool of "systemic" low-level Dpp for additional functions.

Thank you for raising this important point. We realized that our original manuscript was inclined to the idea that BMP ligands are specifically expressed from the hub. We fully agree that BMP ligands can be available outside of the niche. We also agree that it is possible that morphotrap would tether ligands secreted from other cell types. We have corrected those portions and now included suggested possibilities for experimental design of morphotrap as follows:

By expressing MT under control of the *fasIII*Gal4 driver in the *mGL-dpp* homozygous background, we reasoned that we could remove all circulating fraction of Dpp, including fractions secreted from the hub cells or from any other cell types, to prevent its effect outside of the niche, and tether them on hub cell membranes to keep it to signal to stem cells via a contact dependent manner (Figure 2B).

2. The authors show (Fig 3) that trapping Dpp on hub cells results in an increase in "symmetric events" and postulate that trapping Dpp away from SGs might increase rates of dedifferentiation. To test this, they set up a sophisticated experiment (used previously to monitor GSC replenishment in the niche): They deplete the niche of GSC by forced, long - lasting expression of the differentiation factor *bam* and then, after shutting down *bam* expression, they measure dedifferentiation by counting the re-establishment of GSCs in the

niche over a longer time period. However, when carefully looking at the graph of 3H, it is clear that the MT expressing niche starts with significantly more GSC than the controls (time point "post" which I assume refers to the time point directly after the last pulse of bam expression). Thus, this experiment can also be interpreted as follows: Dedifferentiation rates are equal but it takes longer for the controls to catch up as they started with fewer GSC on average. This trend (more GSC at timepoint 0) is also visible in the experiments of Fig 4 (panels G,H) where BMP signal transducers are depleted in SGs. Thus, to me it seems that, for some reason, the niche with trapped Dpp is more resistant to the elimination of GSCs (with the hs-bam tool) than the controls. This would refute the model that the low levels of Dpp in SGs repress dedifferentiation.

Thank you for pointing this out. Similar concern was also raised by Reviewer2. Indeed, we observed higher GSC number at day0 recovery point when BMP signal was suppressed in differentiating germ cells. We have speculated that this could be caused by constant dedifferentiation events even during heat shock treatments. However, it had never been directly tested.

To confirm whether Dpp signal directly signal to differentiating cells to repress dedifferentiation, we have conducted long-term live imaging to monitor dedifferentiation events as suggested by reviewer2. As expected, we observed significant increases in the frequency of dedifferentiation events when Tkv is depleted in differentiating cells (**new Figure S4 and new supplementary video S1**), indicating that the observed quicker recovery is indeed caused by enhancement of dedifferentiation rather than defect in GSC elimination. Relevant portion of the text reads as follows:

To further confirm that the observed faster recovery is surely caused by accelerated de-differentiation, we conducted long-term live imaging of bam>Tkv RNAi flies to measure the frequency of de-differentiation events (Figure S4A-C, supplemental video S1, 2). We monitored the division of GSCs expressing mCherry-Vasa for 16 hours. Strikingly, we found dramatic increase of frequency of de-differentiation events in Tkv RNAi expressing testes, indicating that diffusible fraction of Dpp prevents de-differentiation.

3. The presence of a Smad binding site in the *bam* promoter (Fig 6) is not supported experimentally. It is completely unclear which method (I guess some in silico tool) the authors identified the sequence at -68 as a Mad binding site (the relevant citations rather refer to the previously identified site that is essential for *bam* repression in the GSC (Ref 30, 31)). Similarly, occupation of the identified sequence by Mad (and loss of Mad binding in the mutants) is not presented. Thus, the results of the reporter analyses are, at this point a correlation.

We appreciate for this reviewer for these comments. We focused on +39 site which was previously reported to be responsive for niche BMP signal, and shown to be directly bound by Mad [8, 11]. We also performed in silico analyses of *bam* promoter using MEME suite (<https://meme-suite.org/meme/>) and it detected the -68 site as a consensus binding sequence of Mad. Therefore, we originally included this site for analysis and found that -68 site is also required for suppression of *bam* in GSCs. In the revised manuscript, we focused on only +39 site because we realized that the opposing function of BMP signal can be explained solely by this site. The revised portion of manuscript reads as follows:

Previously, Mad/Med have been shown to directly bind to silencer element within *bam* promoter to suppress its expression [8]. Therefore, we next wondered whether Mad directly act on *bam* promoter both in GSCs and GB/SGs to exert opposite roles. If so, the Mad binding domain in the silencing element of *bam* promoter may alter its function from a repressor to an activator. To test this possibility, we generated *bam* promoter reporter constructs harboring same mutation on +39 site as previously described (Figure 7B) [12]. We observed expression of +39 site-mutated reporter in GSCs, indicating that this site is required for suppression of *bam* expression in GSCs similar to female GSCs [12]. In addition, we noticed that +39 site-mutated-reporter shows drastically lower reporter intensity relative to the control reporter specifically in the 4-8 SGs (Figure 7D-G), indicating that the +39 site is required for upregulation of *bam* in 4-8 SGs opposed to its silencer effect in GSCs, indicating that Mad acts oppositely on +39 site of *bam* promoter in GSCs and in SGs (Figure 7H).

In addition (but I have had difficulties to understand the exact methods here and I might have missed something), I find the construction of the reporters both unconventional and awkward. If I got this right, those are not neutral reporters that would allow for quantitative comparison of an inert marker (GFP) in an otherwise wild-type tested. They do all contain a bamGFP fusion, which when expressed (at different levels and patterns due to the introduced mutations) will also skew the relative cellular composition of the tissue. For example, mutations that result in loss of bam repression in GSC will activate bamGFP in these cells and result in their differentiation and loss. This would make the comparison of the different transgenic reporters very difficult. I would like again to emphasize here that I might have missed something and, hopefully, misinterpreted the experiment!

This reviewer's interpretation is correct. The previous versions of our reporter are containing ORF of bam gene. We once made EGFP reporter without bam ORF in the past, but the signal was too weak to detect. Therefore, we mimicked reporters containing bam ORF exactly as used in the past report [12]. However, we are also aware that this was not the best design to evaluate bam transcription because of the potential secondary effect caused by misexpression of Bam.

In the revised manuscript, we constructed new reporters without the bam ORF. We used mGL which has greater brightness and rapid maturation [13]. Luckily, we have obtained detectable fluorescence of mGL reporters and collected data that is now presented in a revised **new Figure 7 (original Figure 6)**.

Other comments

1. In my opinion, the schematic of 1C (generation of mGL-dpp) makes no sense. The drawing implies that there is some linear intermediate with the given composition, which I do not think is the case. Judging by the description in the legend (the dedicated methods section is not helpful) and the information of Ref20, the recombination event to remove the "last exon of endogenous dpp" is in trans with the PBac(RB) chromosome. Even if there is an

error in my reasoning and I am missing a step, the orientation of the FRTs in the construct would catalyze an inversion and not an excision. Please clarify.

Thank you for noting this mistake. Indeed, the reviewer's interpretation about the event to remove the exon in trans to the chromosome is correct. We have updated **Figure 1C** to accurately reflect our methods.

2. mGL-Dpp is central to this study, however, the distribution of the molecule is puzzling to me. In a previous paper by the same authors (Ref 17) a similar tool to monitor endogenous Dpp (mCherry Dpp) was used. As expected, the majority of mCherryDpp signal was in the hub cells that produce the fusion protein. In this study, this is not the case: The hub cells can be hardly identified as source cells for mGL-Dpp (by enriched Dpp levels) as would be expected. All we see is a uniform, low-level signal with no signs of enrichment in or near the source (hub). What is the explanation for this, and especially the discrepancy to the previous study?

We appreciate this perceptive comment. First, we too noted that the mCherry-Dpp shows more enriched signal in the hub than mGL-Dpp (We have now provided a comparison of all fluorescent tagged lines in **new Figure S1**). It should be noted that the location of mCherry tag was inserted upstream of last processing site of Dpp, thus there is a possibility that non-tagged Dpp and/or free mCherry are present in the tissue. This may potentially cause the different pattern of fluorescent signal in the testis. We have included these points in the revised manuscript (as follows) and also added a table to explain the difference of dpp alleles in **new supplemental Table S1** (below).

Taken together, these data indicate that Dpp likely freely diffuses from the niche and almost uniformly distributes throughout the testis. Supporting this data, we also observed similar distribution pattern of mScarlet-tagged Dpp (*mSC-dpp*) expressed from the endogenous locus allele [14] (Figure S1A, B). In contrast to these alleles, we observed the GFP-Dpp show almost undetectable fluorescent signal, and mCherry-Dpp shows highly concentrated signal within the

hub as we reported previously [15] (Figure S1A, B). It should be noted that the GFP and mCherry tags were inserted into upstream of last processing site of Dpp, thus there is a possibility that non-tagged Dpp and/or free fluorescent proteins are present in the tissue. This may potentially cause the different pattern of fluorescent signal in the testis (Table S1).

Alleles	mCherry-dpp * ¹	GFP-dpp * ¹	mGL-dpp * ¹	mSC-dpp * ¹	HA-dpp * ¹	GFP-dpp
Sources	Fereres et al., 2019	Gift from Thomas Kornberg	This study, Rasouliha et al., 2023	This study, Rasouliha et al., 2023	Matsuda et al., 2021	Matsuda et al, 2021
Phenotypes	Homozygous viable Potential generation of non-tagged Dpp fraction* ²	Homozygous viable Potential generation of non-tagged Dpp fraction* ²	Homozygous semi-lethal Rescuable with pPA dpp 8391/X	Homozygous semi-lethal Rescuable with pPA dpp 8391/X	None	Haploinsufficient Partially rescuable with pPA dpp 8391/X (patterning defects)
Tag location	AA465	AA465	AA485	AA485	AA485	AA485
GSC phenotypes after expression of MT under hub driver (FasIIIGal4)	Not tested	Reduced pMad with mCD8-MT	Reduced pMad with mCD8-MT	Not tested	Reduced pMad with HAtrap	Not tested
De-differentiation phenotypes after expression of MT under hub driver (FasIIIGal4)	Not tested	Accelerated dedifferentiation with Nrv-MT trap	Accelerated dedifferentiation with Nrv-MT trap	Not tested	Not tested	Not tested

*¹ Alleles used in this study

*² Tags placed at AA465 may be cut out at the last furin processing site

Also, the model suggested by the authors poses that there are two regions with different Dpp signaling levels in the signal receiving cells: high levels at the GSC region and low at the GB/SG area but this is hardly reflected in the distribution of mGL-Dpp.

We agree with this reviewer's view that Dpp distribution in the testis is uniform. We discussed several possible reasons why strong pMad can be seen exclusively in GSCs, which reads as follows:

The niche Dpp has been postulated to act as a highly localized signal as pMad can be observed exclusively in GSCs and their only immediate progenies. However, our work suggests that the distribution of Dpp is rather uniform, implying that the signal reception is not uniform. Indeed, many works have revealed how a steep gradient of BMP response is established within just one cell diameter [16-27]. Many of these studies postulated redundant mechanisms in which either stem cells enhance the signal reception, or differentiating cells actively suppress it. Alternatively, specific composition of ligands may not be so uniform. Our study suggests a requirement of both Dpp and Gbb for observed cellular responses, and co-localization of Dpp-Gbb was exclusively seen in the hub (Figure 5L, M), suggesting a possibility that Dpp-Gbb may form heterodimer and distribution of which might be tightly restricted around the hub even though each ligand alone can diffuse freely. A recent report has demonstrated that the heterodimers of Dpp-Gbb preferentially produced in Dpp producing cells and play critical roles during development [1, 28]. This may contribute to the formation of sharply graded signaling outcomes around the niche. It would be interesting to investigate the precise distribution and composition of these ligands in a quantitative manner.

3. The expression of the morphotrap (MT) results in elimination of mGL signal outside the niche, however it is odd that enrichment of the signal at the membranes of hub cells expressing the MTs is undetectable. This is in contrast to the strong enrichment of mGL signal on membranes of germline cells (nosGal4). What is the reason for this?

We appreciate this important comment. Similar concern was made by all of the reviewers. Indeed, we found that nrv-MT expression does show tethered mGL-Dpp (**new Figure 2H**), the signal intensity was not so high reflecting all circulating Dpp fractions. We found evidence that trapped mGL-Dpp may be constantly internalized in hub cells and degraded. However, because GSCs can still receive signal from tethered Dpp without impact on the

strength of signalling, as evidenced by our observation that the pMad level is equivalent to the control, we conclude that this condition is suitable for our experimental design (see details in our response to reviewer1's "Validity of Data" section).

4. In connection to the above: The main driver in this study is the newly generated mGLDpp, which seems to be an excellent tool also for other contexts. Given its central role, it is essential to characterize this allele carefully and provide the data (the statement in the methods section "detailed characterization will be reported elsewhere" is not satisfactory).

Also, the authors claim that the construct is "semilethal" and flies homozygous for this chromosome need to be supplemented with a transgenic construct that provides Dpp in the embryo. This is somehow at odds with the statement in the methods (line 385) that mGL-dpp homozygous flies show no obvious phenotypes. Do the authors refer to escapers (as there seems to be a fraction that bypasses embryonic lethality without the help of the transgenic Dpp source)?

We agree that description of mGL-Dpp allele in the method section was unclear and not sufficient. We have posted new preprint using mGL-Dpp in wing discs, which shows additional characterization of this allele: bioRxiv 2023.03.27.534445; doi: <https://doi.org/10.1101/2023.03.27.534445>.

The relevant text in the revised manuscript reads as follows:

The detail procedure to generate endogenously tagged *dpp* alleles were previously reported [29]. In brief, utilizing the *attP* sites in a MiMIC transposon inserted in the *dpp* locus (MiMIC *dppMI03752*, BDSC36399), about 4.4 kb of the *dpp* genomic sequences containing the second (last) coding exon of *dpp* including a tag and its flanking sequences was inserted in the intron between *dpp*'s two coding exons. The endogenous exon was then removed using FLP-FRT to keep only the tagged exon. mGL (mGreenLantern [13]) or mSC (mScarlet [30]) were inserted in frame after amino acid 485 (NM_164488.2) after the last processing site to tag all the Dpp

mature ligands. mGL coding sequences after the last processing site. The detail characterization of these alleles are described in [14].

Along these lines, it is important to show that the additional transgenic dpp source is not active during spermatogenesis as this would obscure the findings. I understand that this might be difficult, as it would require some dpp allele that specifically affects GSC maintenance to demonstrate that it cannot be restored by the transgenic Dpp source. Alternatively, the authors might be able to address this by quantitatively showing that homozygous “escapers” males do have normal spermatogenesis (i.e. the transgenic dpp source does not provide ligands during spermatogenesis).

We appreciate the suggestion to rule out the effects of the transgenic dpp source during spermatogenesis. Similar concern was also raised by Reviewer2.

To rescue semi-lethality of homozygous mGL-dpp allele, we used a transgene *pPA dpp 8391/X*, which contains the genomic region of dpp critical for the early embryogenesis [2] but does not rescue the wing phenotypes of dpp mutants [3]. To test if this transgene is not working in the testis to modify mGL-dpp trap phenotype, we expressed GFP RNAi under the hub driver (*FasIII-Gal4*) in the *pPA dpp 8391/X; mGL-dpp/mGL-dpp* background to specifically knock down mGL-dpp (but not dpp on *pPA dpp 8391/X*). We found that pMad signal is significantly reduced in GSCs, indicating that *pPA dpp 8391/X* transgene is unlikely expressed in the hub. These data are now presented in **new Figure S2H**, and we explain the point as below:

“It should be noted that knock-down of mGL-tagged Dpp in the hub (*pPA dpp 8391/X, fasIII>GFP-IR, mGL-dpp/mGL-dpp*) significantly reduced pMad signal in GSCs, confirming that rescuing transgene (*pPA dpp 8391/X*) is non-functional in the hub (Figure S2H).”

5. The dextran experiment is confusing: The authors state (starting at line 92) that they subjected testes from mGL-Dpp flies to Dextran incubation. Thus, I would expect to see two panels of the very same tissue/SG area in two different channels (mGLDpp and 10kD-Dextran647) to demonstrate co-localization. Instead, the authors show different tissues and claim that the pattern is similar (i.e. panel E and F are not the same sample and the same is true for panel H and I). Why not monitoring both mGLDpp and 10kD-Dextran647 in the same sample? In addition, the images are not very convincing. Is it possible to use an extracellular staining protocol to exclusively access extracellular fractions of Dpp?

Thank you for pointing this out. We apologize that the original description was incorrect, as we originally used wildtype testis for 10kDa-Dextran647 treatment and compared the pattern with mGL-Dpp. We have corrected this in the revised manuscript, where we present data from imaging of mGL-Dpp with Dextran647 in the same sample (**new Figure 1H**). We also have updated the images in **Figure 1D-H**.

We have attempted but have not succeeded for extracellular staining protocol due to high non-specific binding of antibodies to the extracellular spaces.

Minor remarks

In general, the draft contains some unclear/clumsy formulations and syntactical flaws and would benefit from a thorough proofreading. Here some examples.

1. The authors use the term "Dpp ligand" (for example line 78) or "Dpp" synonymously throughout the paper. Please use consistently "Dpp" when referring to the ligand. The term "Dpp ligand" is confusing because it can be interpreted as "the ligand of Dpp" implying that Dpp is the receptor.

2. Line 42 : Please specify that the paragraph describes the male germ line stem cell niche

3. Sentence starting at line 92. Please use “we incubated ... in media with freely ..., which is a of similar size...”

4. Introduce pMad (“Phosphorylated Mad (pMad) ..., a read out of Dpp signal activation”) earlier: In line 132 instead of 142.

5. Confusing phrasing in paragraph of line 157: The dedifferentiation event in Fig 3A-2 is not a subclass of symmetric GSC division. Change “division” of line 159 with “event”?

6. Fig. 3H: define “post” in the graph. If it is directly after the last heat shock, then it would be better to define it as “0 Days” in recovery

Thank you for finding these errors. These were very helpful. We corrected comments 1-6 as suggested.

7. Please explain what we see in Fig 4 A and B. I guess the loss of GSCs in B is monitored by the absence of fusome-containing cells ?

We judged the loss of GSCs by monitoring absence of Vasa positive cells around the hub. We have now added the information in the figure legend.

8. Please display single channels for bam levels (Fig 5A-C) as the slight differences are masked by the DAPI channel.

We have updated **Figures 6A-D (originally 5A-C)** to show separate channels and added **new Figure S6** to show details of the bam expression pattern.

We would like to note that we made a correction on interpretation of the data with which we originally stated that the opposed effect of Mad occurs in a concentration dependent manner. This statement was based on the data shown in **Figure 7A (originally Figure 6C)** in which Mad overexpression and Mad RNAi have opposite effect on de-differentiation. However, we later realized that this data rather disagrees with the idea of “concentration dependency” because the effect on dedifferentiation (low-Mad condition) was strengthened by overexpressing Mad (high-Mad condition), and it does not “flip” the effect. Although these results may not be enough to completely deny concentration dependency, we have suggested likelihood of involvement of other factors to conclude this section in the revised manuscript. We corrected model figure, **Figure 7H** as well as our description as below:

In the testis, Dpp signal is highest in GSCs, with strong pMad intensity, and substantially lower in GB/SGs (Figure 2I). Therefore, we wondered if the opposing Dpp functional outputs depend on pMad concentration. To mimic GSC’s pMad level in differentiating cells, we overexpressed Mad (bam>Mad). If high concentration of Mad acts oppositely to a low concentration of Mad, we should observe similar effect between Mad RNAi and Mad overexpression. However, we found that Mad overexpression in hs-Bam flies cause a reduction of recovery rate of GSCs after heat-shock mediated GSC depletion (Figure 7A), opposite of the effect of bam>Mad RNAi, suggesting that functional switch of Mad from repressor to activator is unlikely dependent on concentration of pMad, and other factor(s) may be involved in this process.

1. Bauer M, Aguilar G, Wharton KA, Matsuda S, Affolter M. Heterodimerization-dependent secretion of bone morphogenetic proteins in Drosophila. *Developmental Cell*. 2023;58(8):645-59.e4. doi: <https://doi.org/10.1016/j.devcel.2023.03.008>.
2. Hoffmann FM, Goodman W. Identification in transgenic animals of the Drosophila decapentaplegic sequences required for embryonic dorsal pattern formation. *Genes Dev*. 1987;1(6):615-25. Epub 1987/08/01. doi: 10.1101/gad.1.6.615. PubMed PMID: 2824286.
3. Simon N, Safyan A, Pyrowolakis G, Matsuda S. Dally is not essential for Dpp spreading or internalization but for Dpp stability by antagonizing Tkv-mediated Dpp internalization. *eLife Sciences Publications, Ltd*; 2023.
4. Herrera SC, Bach EA. JNK signaling triggers spermatogonial dedifferentiation during chronic stress to maintain the germline stem cell pool in the Drosophila testis. *eLife*. 2018;7:e36095. doi: 10.7554/eLife.36095.
5. Sheng XR, Matunis E. Live imaging of the Drosophila spermatogonial stem cell niche reveals novel mechanisms regulating germline stem cell output. *Development*. 2011;138(16):3367-76. Epub

2011/07/15. doi: 10.1242/dev.065797. PubMed PMID: 21752931; PubMed Central PMCID: PMC3143561.

6. Insko ML, Leon A, Tam CH, McKearin DM, Fuller MT. Accumulation of a differentiation regulator specifies transit amplifying division number in an adult stem cell lineage. *Proc Natl Acad Sci U S A*. 2009;106(52):22311-6. Epub 2009/12/19. doi: 10.1073/pnas.0912454106. PubMed PMID: 20018708; PubMed Central PMCID: PMC2799733.
7. Li Y, Minor NT, Park JK, McKearin DM, Maines JZ. Bam and Bgcn antagonize Nanos-dependent germ-line stem cell maintenance. *Proceedings of the National Academy of Sciences*. 2009;106(23):9304-9. doi: 10.1073/pnas.0901452106.
8. Chen D, McKearin D. Dpp Signaling Silences bam Transcription Directly to Establish Asymmetric Divisions of Germline Stem Cells. *Current Biology*. 2003;13(20):1786-91. doi: <https://doi.org/10.1016/j.cub.2003.09.033>.
9. Schulz C, Kiger AA, Tazuke SI, Yamashita YM, Pantalena-Filho LC, Jones DL, et al. A misexpression screen reveals effects of bag-of-marbles and TGF beta class signaling on the Drosophila male germ-line stem cell lineage. *Genetics*. 2004;167(2):707-23. Epub 2004/07/09. doi: 10.1534/genetics.103.023184. PubMed PMID: 15238523; PubMed Central PMCID: PMC1470893.
10. Kawase E, Wong MD, Ding BC, Xie T. Gbb/Bmp signaling is essential for maintaining germline stem cells and for repressing bam transcription in the Drosophilatestis. *Development*. 2004;131(6):1365-75. doi: 10.1242/dev.01025.
11. Chen D, McKearin DM. A discrete transcriptional silencer in the bam gene determines asymmetric division of the Drosophila germline stem cell. *Development*. 2003;130(6):1159-70. Epub 2003/02/07. doi: 10.1242/dev.00325. PubMed PMID: 12571107.
12. Chen D, McKearin DM. A discrete transcriptional silencer in the bam gene determines asymmetric division of the Drosophila germline stem cell. *Development*. 2003;130(6):1159-70. doi: 10.1242/dev.00325.
13. Campbell BC, Nabel EM, Murdock MH, Lao-Peregrin C, Tsoulfas P, Blackmore MG, et al. mGreenLantern: a bright monomeric fluorescent protein with rapid expression and cell filling properties for neuronal imaging. *Proceedings of the National Academy of Sciences*. 2020;117(48):30710-21. doi: 10.1073/pnas.2000942117.
14. Sheida Hadji R, Gustavo A, Cindy R, Shinya M. Shaping and interpretation of Dpp morphogen gradient by endocytic trafficking. *bioRxiv*. 2023:2023.03.27.534445. doi: 10.1101/2023.03.27.534445.
15. Ladyzhets S, Antel M, Simao T, Gasek N, Cowan AE, Inaba M. Self-limiting stem-cell niche signaling through degradation of a stem-cell receptor. *PLOS Biology*. 2020;18(12):e3001003. doi: 10.1371/journal.pbio.3001003.
16. Guo Z, Wang Z. The glypican Dally is required in the niche for the maintenance of germline stem cells and short-range BMP signaling in the Drosophila ovary. *Development*. 2009;136(21):3627-35. Epub 2009/10/02. doi: 10.1242/dev.036939. PubMed PMID: 19793889.
17. Liu M, Lim TM, Cai Y. The Drosophila female germline stem cell lineage acts to spatially restrict DPP function within the niche. *Sci Signal*. 2010;3(132):ra57. Epub 2010/07/29. doi: 10.1126/scisignal.2000740. PubMed PMID: 20664066.
18. Harris RE, Pargett M, Sutcliffe C, Umulis D, Ashe HL. Brat promotes stem cell differentiation via control of a bistable switch that restricts BMP signaling. *Dev Cell*. 2011;20(1):72-83. Epub 2011/01/18. doi: 10.1016/j.devcel.2010.11.019. PubMed PMID: 21238926; PubMed Central PMCID: PMC3178012.
19. Van De Bor V, Zimniak G, Papone L, Cerezo D, Malbouyres M, Juan T, et al. Companion Blood Cells Control Ovarian Stem Cell Niche Microenvironment and Homeostasis. *Cell Rep*. 2015;13(3):546-60. Epub 2015/10/13. doi: 10.1016/j.celrep.2015.09.008. PubMed PMID: 26456819.

20. Wang X, Harris RE, Bayston LJ, Ashe HL. Type IV collagens regulate BMP signalling in *Drosophila*. *Nature*. 2008;455(7209):72-7. Epub 2008/08/15. doi: 10.1038/nature07214. PubMed PMID: 18701888.
21. Xia L, Zheng X, Zheng W, Zhang G, Wang H, Tao Y, et al. The niche-dependent feedback loop generates a BMP activity gradient to determine the germline stem cell fate. *Curr Biol*. 2012;22(6):515-21. Epub 2012/03/01. doi: 10.1016/j.cub.2012.01.056. PubMed PMID: 22365848.
22. Eliazer S, Palacios V, Wang Z, Kollipara RK, Kittler R, Buszczak M. Lsd1 restricts the number of germline stem cells by regulating multiple targets in escort cells. *PLoS Genet*. 2014;10(3):e1004200. Epub 2014/03/15. doi: 10.1371/journal.pgen.1004200. PubMed PMID: 24625679; PubMed Central PMCID: PMC3952827.
23. Tseng CY, Su YH, Yang SM, Lin KY, Lai CM, Rastegari E, et al. Smad-Independent BMP Signaling in Somatic Cells Limits the Size of the Germline Stem Cell Pool. *Stem Cell Reports*. 2018;11(3):811-27. Epub 2018/08/21. doi: 10.1016/j.stemcr.2018.07.008. PubMed PMID: 30122445; PubMed Central PMCID: PMC6135924.
24. Jiang X, Xia L, Chen D, Yang Y, Huang H, Yang L, et al. Otefin, a nuclear membrane protein, determines the fate of germline stem cells in *Drosophila* via interaction with Smad complexes. *Dev Cell*. 2008;14(4):494-506. Epub 2008/04/16. doi: 10.1016/j.devcel.2008.02.018. PubMed PMID: 18410727.
25. Xia L, Jia S, Huang S, Wang H, Zhu Y, Mu Y, et al. The Fused/Smurf complex controls the fate of *Drosophila* germline stem cells by generating a gradient BMP response. *Cell*. 2010;143(6):978-90. Epub 2010/12/15. doi: 10.1016/j.cell.2010.11.022. PubMed PMID: 21145463.
26. Schulz C, Wood CG, Jones DL, Tazuke SI, Fuller MT. Signaling from germ cells mediated by the rhomboid homolog *stet* organizes encapsulation by somatic support cells. *Development*. 2002;129(19):4523-34. Epub 2002/09/12. doi: 10.1242/dev.129.19.4523. PubMed PMID: 12223409.
27. Sardi J, Bener MB, Simao T, Descoteaux AE, Slepchenko BM, Inaba M. Mad dephosphorylation at the nuclear pore is essential for asymmetric stem cell division. *Proceedings of the National Academy of Sciences*. 2021;118(13):e2006786118. doi: doi:10.1073/pnas.2006786118.
28. Matsuda S, Shimmi O. Directional transport and active retention of Dpp/BMP create wing vein patterns in *Drosophila*. *Developmental Biology*. 2012;366(2):153-62. doi: <https://doi.org/10.1016/j.ydbio.2012.04.009>.
29. Matsuda S, Schaefer JV, Mii Y, Hori Y, Bieli D, Taira M, et al. Asymmetric requirement of Dpp/BMP morphogen dispersal in the *Drosophila* wing disc. *Nature Communications*. 2021;12(1):6435. doi: 10.1038/s41467-021-26726-6.
30. Bindels DS, Haarbosch L, van Weeren L, Postma M, Wiese KE, Mastop M, et al. mScarlet: a bright monomeric red fluorescent protein for cellular imaging. *Nat Methods*. 2017;14(1):53-6. doi: 10.1038/nmeth.4074.

REVIEWER COMMENTS

Reviewer #1 (Remarks to the Author):

This reviewer is satisfied with the changes that have been made to this already strong manuscript. Some specific comments on aspects of my initial review that were addressed by the authors are below.

Validity of Data

One of my initial comments addressed the point that if the MT construct was tethering the diffusible fraction of dpp to the niche region, that we should be able to detect higher fluorescence of dpp-GFP in the niche region. The authors address this comment by showing that while they do not, in fact, see higher dpp-GFP at the niche when tethered, a likely explanation could be the high levels of lysosomal degradation of the Dpp-mGL signal.

A minor thought that arises here is that it would be nice to see the increase of internalized mGL-Dpp quantified. For example, the most central cells in control hubs appear to have a high frequency of internalized lysosomal signal that is similar the frequency of internalized signal throughout the entire hub when MT is used to tether the signal. Is it true that more peripheral hub cells now begin to internalize signal when MT is expressed, accounting for decreased signal on the surface of hub cells?

Methods

Full p-values are included, as is the catalog number for the p-Mad primary antibody used in this study. These additions address some of my previous concerns.

I also requested that the authors provide additional information about their live imaging protocol, for full reproducibility. In this revised version, size of Z stacks is reported, and time intervals are reported for the long-term live imaging. Frequency of time intervals for short term live imaging is not reported. If timelapses were acquired, knowing time intervals helps the reader to understand the dynamics captured here. However, if short term live imaging is not meant to capture cell dynamics over time, then perhaps timelapses were not acquired.

Reviewer #2 (Remarks to the Author):

Upon revision, the authors have remarkably addressed all issues pointed by this reviewer, and I find it quite acceptable for publication.

Reviewer #3 (Remarks to the Author):

The authors provide a new version of the MS, which contains new experiments and addresses some of the points raised by the referees. However, I fell that some point have not been addressed appropriately and some new points, in connection with the new experiments, arise.

Main points

1. I find the part describing the effects on bam expression (Fig 6 and 7) particularly difficult to follow and flawed, for the following reasons:

In the previous version, the authors could show that bam levels were reduced whenever the “diffusible fraction of Dpp” was eliminated (see rather convincing Fig. 5 of the previous version of the MS). Here the authors seem to see a different phenotype: levels of bam are not affected, but it is rather the spatial distribution of bam that changes, with peak levels being reached only at later stages (16 cell cysts). Is there a reason for this discrepancy? The tools and genotypes are identical.... My statement refers to the two first panels of Fig. 6. As for the other panels of the figure I cannot really judge, as they are either not referred/described in the main body text, or mislabeled and swapped to such a degree that I simply cannot follow.

Similar problems arise with Figure 7 and the corresponding main body paragraphs: First of all the sentence: “If high concentration of Mad acts oppositely to a low concentration of Mad, we should observe similar effect between Mad RNAi and Mad overexpression.” (absent in the first version of the MS) makes no sense to me. Then, based on the outcome of the dedifferentiation assay, they decide that the differential effects of pMad in GSC and differentiating cells are not due to the concentration but rather depends on other factors. This is quite the opposite of what was the major and main claim of the first version, based on exactly the same experiments. The logic of the current interpretation – and especially the change in the interpretation between the two versions escapes me completely and is very confusing and problematic. The previous version on the differential effects of pMad was based on the collaboration between two (one previously characterized and one potential) pMad binding sites. The model would be elegant and was a main “selling point” in the previous version as it could explain opposing transcriptional outcome based on concentration-dependent engagement of the two sites. Here, the second pMad site is not mentioned any more and the (mainly same) results are interpreted completely different. Specifically, the authors simply state that both effects of Dpp/Gbb are mediated by the same (already characterized) Mad/Medea binding site in the promoter of bam and that the different transcriptional outcome (full repression in GSC versus slight enhancement of expression) depends on mechanisms and factors that are unclear- thus not adding to our understanding. In addition, while the genetic analyses of Fig 6 and the promoter analyses of Fig 7 seem not to align: while the results of Fig 6 suggest that the diffusible fraction is controlling the timing of bam expression, the results of Fig 7 focus on the expression levels.

2. Discrepancies in Figure labels and annotations, missing description as well as multiple flaws in wording and syntax make the MS very hard to read and comprehend, especially for a non-specialist. Some of these points are listed under “minor remarks”, but this is just a very small fraction. The MS requires extensive and careful rewriting to reach the level required for publication.

Other points

1. In answering my question on whether the transgenic Dpp construct used to overcome embryonic lethality might somehow contribute Dpp ligands during spermatogenesis (thus complicating the analyses) they authors provide an additional experiment (described in line 169 and Fig S2H). Specifically, they show that depleting Dpp in the hub reduces pMad levels in GSC in the presence of the transgenic construct. In my understanding, unless the effect is absolute (i.e. complete loss of pMad), this cannot disprove the contribution of ligands for the transgene. Similarly, since the authors agree that there are other sources of Dpp besides hub-derived Dpp, this experiment is not even adequate to define the hub as the main Dpp source.

2. While the authors in the rebuttal letter agree that there are other sources of Dpp than the hub (see also above), the narrative of the papers still suggest exactly this. For example, the authors use the term long-range diffusion/distribution of Dpp, however since Dpp might come from cells adjacent to the differentiating cells (and not the hub), the diffusion might not be that long after all. Another example : even the schematic in Fig 2A suggest that the MTs trap Dpp that is derived from the same cells (hub) that also present the MT. I believe this needs to be clarified.

Minor suggestions

Line 83:

“hence we introduced a transgenic allele (pPA dpp 8391/X) containing genomic region of dpp expressed only in embryonic stage to assist embryonic development.”

Awkward phrasing.

Figure 1:

The figure has been "enriched" by a panel showing GFP-Dpp (panel F), however the panel (and the outcome of this experiment) are not explicitly discussed. Is this the allele from the Kornberg lab? If so (table 1 suggests that this is the only GFP-Dpp used), it seems that the distribution between cells is even better visible with this tool (compare 1E and 1F). Why not using this allele? Also, while the table 1 is a good attempt to provide clarity over the available and used alleles, it would be good to use consistent nomenclature throughout the draft to make the life of readers (and reviewers) easy. For example, I cannot easily allocate the " (Crispr knock in)" alleles of Suppl Fig. 1.

Line 112-118:

Fix syntax and grammar. All three sentences are flawed.

Line 149:

"Indeed, the Dpp-mGL signal in hub lysosomes after 4-hour chloroquine treatment showed increase in Dpp-mGL in hub cells of MT expressing testes!"

please use

Paragraph starting with line 152:

All figure/panel annotations are wrong, including the suppl. Figure. Fix!

"Indeed, the Dpp-mGL signal in hub lysosomes after 4-hour chloroquine treatment was increased in MT expressing testes"

Line 157:

"In comparison, mGL-Dpp, fasIII>Nrv-MT" replace with "In contrast, mGL-Dpp, fasIII>Nrv-MT"

Paragraph starting with line:

All figure/panel annotations are wrong, Fix!

Also legend for panels N,O misleading : N,O are not the of the same genotype as M, only O is (I hope)

Line 276:

"Next, we conducted trapped HA-tagged Dpp or Gbb using HA-trap" Unclear, do you mean "Next, we trapped HA-tagged Dpp..." ? Would be also good to introduce the structure of the trapping system (as you did for MT)

First of all, we thank editor and the reviewers for giving us critical suggestions on our original and revised manuscripts. Thankfully, current version has been improved significantly strengthened our conclusions. Below is the response for reviewers' comments. Reviewer comments are blue, our responses are black.

Reviewer #1 (Remarks to the Author):

This reviewer is satisfied with the changes that have been made to this already strong manuscript. Some specific comments on aspects of my initial review that were addressed by the authors are below.

Validity of Data

One of my initial comments addressed the point that if the MT construct was tethering the diffusible fraction of dpp to the niche region, that we should be able to detect higher fluorescence of dpp-GFP in the niche region. The authors address this comment by showing that while they do not, in fact, see higher dpp-GFP at the niche when tethered, a likely explanation could be the high levels of lysosomal degradation of the Dpp-mGL signal.

A minor thought that arises here is that it would be nice to see the increase of internalized mGL-Dpp quantified. For example, the most central cells in control hubs appear to have a high frequency of internalized lysosomal signal that is similar the frequency of internalized signal throughout the entire hub when MT is used to tether the signal. Is it true that more peripheral hub cells now begin to internalize signal when MT is expressed, accounting for decreased signal on the surface of hub cells?

Thank you for bringing up this interesting possibility. We looked at internalized Dpp signal more carefully but did not detect any location bias of internalized Dpp throughout the hub. This indicated that all hub cells similarly secrete Dpp and there is unlikely the mechanism of directional secretion. While we decided not to describe this point as it remains speculative, it will be an exciting future study to investigate the mechanisms of Dpp secretion from the niche.

Methods

Full p-values are included, as is the catalog number for the p-Mad primary antibody used in this study. These additions address some of my previous concerns.

I also requested that the authors provide additional information about their live imaging protocol, for full reproducibility. In this revised version, size of Z stacks is reported, and time intervals are reported for the long-term live imaging. Frequency of time intervals for short term live imaging is not reported. If timelapses were acquired, knowing time intervals helps the reader to understand the dynamics captured here. However, if short term live imaging is not meant to capture cell dynamics over time, then perhaps timelapses were not acquired.

Thank you again for your comment on this issue. After reading this comment, we realized that we did not state the rationale to use short term live imaging for static image acquisition. We used this method to avoid potential loss of fluorescent signal or tagged protein itself located in extracellular space by fixation and permeabilization. We did not intend to capture dynamics over time and did not use time-lapse acquisition. Now these points are added in the method section and method used for each representative image and quantification (short-term live imaging vs. fixed staining) are in the figure legends.

Reviewer #2 (Remarks to the Author):

Upon revision, the authors have remarkably addressed all issues pointed by this reviewer, and I find it quite acceptable for publication.

Reviewer #3 (Remarks to the Author):

The authors provide a new version of the MS, which contains new experiments and addresses some of the points raised by the referees. However, I felt that some point have not been addressed appropriately and some new points, in connection with the new experiments, arise.

We appreciate for this reviewer for carefully reading our previous versions of manuscript and his/her insightful comments.

Main points

1. I find the part describing the effects on bam expression (Fig 6 and 7) particularly difficult to follow and flawed, for the following reasons:

In the previous version, the authors could show that bam levels were reduced whenever the “diffusible fraction of Dpp” was eliminated (see rather convincing Fig. 5 of the previous version of the MS). Here the authors seem to see a different phenotype: levels of bam are not affected, but it is rather the spatial distribution of bam that changes, with peak levels being reached only at later stages (16 cell cysts). Is there a reason for this discrepancy? The tools and genotypes are identical.... My statement refers to the two first panels of Fig. 6. As for the other panels of the figure I cannot really judge, as they are either not referred/described in the main body text, or mislabeled and swapped to such a degree that I simply cannot follow.

We clearly see this confusion due to the change we made in describing diffusing Dpp effect on bam expression pattern. First, we would like to emphasize that we did not mean to change

initial statement that BMP mutants have lower Bam level in 4-8 SGs (we reposted the quantification for 4-8 SGs).

During the previous revision cycle, we also scored Bam levels in the 16-cell SG stage as suggested by reviewer #2 because the insufficient level of Bam in 16-SGs could cause extra division of SGs. Based on this scoring, we realized that Bam levels catch up in 16 SG stages in all BMP signal mutant genotypes we examined including we have shown in FigS6 (we apologize for missing citation of FigS6. We added a description in the newest version).

In addition, while the genetic analyses of Fig 6 and the promoter analyses of Fig 7 seem not to align: while the results of Fig 6 suggest that the diffusible fraction is controlling the timing of bam expression, the results of Fig 7 focus on the expression levels.

We indeed realized that the bam mutant reporter pattern and bam staining pattern in mutant fly for BMP signal components do not align as pointed out above. Since our bam-mGL reporters only reflect transcriptional regulation, we speculate that how BMP signal mutant catch up Bam level later in 16-SGs might be via post-transcriptional feedback mechanisms. We added discussion of this point as follows:

“It should be noted that all bam-mGL reporters show uniform expression levels throughout all SG stages (4-to-16 SGs), unlike the endogenous Bam pattern where Bam staining shows peak intensity at 8-cell SG stage. This indicates that downregulation of am in 16-SGs is post-transcriptionally regulated. Therefore, we speculate that the observed shift of bam expression peak from 4-to-8 cell SGs to 16-cell SGs in BMP mutants (Figure 6A-F) is likely caused by a post-transcriptional feedback mechanism in response to reduction of bam transcription in earlier SGs. Further studies will be necessary to fully understand this feedback mechanism.”

Similar problems arise with Figure 7 and the corresponding main body paragraphs: First of all the sentence: “If high concentration of Mad acts oppositely to a low concentration of Mad, we should observe similar effect between Mad RNAi and Mad overexpression.” (absent in the first version of the MS) makes no sense to me.

Then, based on the outcome of the dedifferentiation assay, they decide that the differential effects of pMad in GSC and differentiating cells are not due to the concentration but rather depends on other factors. This is quite the opposite of what was the major and main claim of the first version, based on exactly the same experiments. The logic of the current interpretation – and especially the change in the interpretation between the two versions escapes me completely and is very confusing and problematic.

The previous version on the differential effects of pMad was based on the collaboration between two (one previously characterized and one potential) pMad binding sites. The model would be elegant and was a main “selling point” in the previous version as it could explain opposing transcriptional outcome based on concentration-dependent engagement of the two sites. Here, the second pMad site is not mentioned any more and the (mainly same) results are interpreted completely different. Specifically, the authors simply state that both effects of

Dpp/Gbb are mediated by the same (already characterized) Mad/Medea binding site in the promoter of *bam* and that the different transcriptional outcome (full repression in GSC versus slight enhancement of expression) depends on mechanisms and factors that are unclear- thus not adding to our understanding.

We apologize again our misinterpretation of the data we made in our original version. Below, we explain the problem again.

Outline of the problem

1. We stated in our very 1st version that Mad regulates *bam* expression oppositely in a concentration dependent manner (i.e., High Mad suppress Bam expression, Low Mad activates Bam expression).
2. However, we realized that our original interpretation was wrong (see more explanation below).
3. Since we formulated the logic behind the Bam reporter experiments based on the pMad concentration effects, the nuance of the following result section also changed significantly between 1st and 2nd versions.

Details of data misinterpretation

Based on the data shown in original Figure 6C (right panel, same as Figure 7A in 1st revision), we have erroneously come to the opposite conclusion than what we should have. Here we more clearly explain our logic and how we see the results fitting into our model.

To mimic GSCs' high pMad level in differentiating cells, we overexpressed Mad under the *bamGal4* driver (*bam>Mad*). If a high concentration of Mad has an opposing function to a low concentration of Mad, we should observe Mad overexpression having an opposing effect on dedifferentiation (i.e, enhancing dedifferentiation, instead of inhibiting). In this case, Mad RNAi and overexpression should show the same outcome. However, we found that Mad overexpression was different than the result of Mad RNAi. Therefore, we concluded that the functional switch of Mad from repressor to activator is unlikely dependent on pMad concentration, at least based on this experiment.

New results support concentration dependency

After we had more careful consideration/discussion on this point, we realized that the previous experiment was not conclusive enough to deny concentration dependency, because

overexpressed Mad is not necessarily phosphorylated and may not faithfully mimic GSCs' pMad level. To assesses pMad concentration effects more directly, we designed a better method by taking advantage of expression of a constitutively active Tkv receptor (Tkv-CA).

Expression of Tkv-CA causes SG-like tumors filled with either Bam positive or Bam negative SG-like populations. We stained these tumor cells with pMad and examined the correlation of pMad levels and bam expression levels (bam-mGL reporter intensity). Using this system, we assessed relationships between pMad concentration and Bam expression (see below).

“In the testis, Dpp signal is highest in GSCs, with strong pMad intensity as compared to GB/SGs (Figure 2I). Therefore, we wondered if the opposing outcomes of Dpp signal depend on pMad concentration. To test the likelihood of this possibility, we took the advantage of nos>TkvCA tumor cells, which express various levels of Bam, to examine relationships between pMad concentration and Bam expression level (Figure 7A). Interestingly, we indeed found that the pMad levels and Bam levels show a non-linear correlation, such that populations with the highest pMad levels have low levels of Bam, whilst populations with modest pMad levels show the highest levels of Bam (Figure 7A, B). These two populations mimic GSCs (high pMad, low or no Bam) and SGs (low pMad, high Bam), suggesting that the opposing outcomes of Dpp signaling are dependent on pMad concentration.”

Reformulated result section without fitting to particular idea

In our previous versions, the description of the Bam reporter results was inclined to the idea of either “concentration dependent” or “concentration independent” models. After careful consideration, we realized that these two models do not necessarily mutually exclusive, and we should not exclude the possibility of involvement of other factors. Therefore, we reformulated result section describing bam reporter without fitting to one of these two models. Now we

included data from both reporters (mutated either -68 and +39 sites or both) and simply described what we saw. Corresponding result section reads as follows:

“The Mad binding domain in the bam promoter has been well characterized in female GSCs (34, 38). In addition to a previously-characterized Mad binding site required for silencing bam in female GSCs (position +39 from the transcription start site, TSS), another putative Mad binding site was reported at position -68 (Figure 7C) (34, 39). To examine the function of these two Mad binding sites in male GSCs and differentiating cells, we generated flies carrying bam promoter reporter constructs with or without mutations at these sites that would abrogate Mad binding (Figure 7C) (34, 38). As reported in female GSCs, we found that the +39 site, but not the -68 site, is required to suppress Bam expression in GSCs, as mutations of +39 site caused precocious expression of the bam reporter in GSCs (Figure 7D, E). Furthermore, we noticed that the +39 mutant reporter showed drastically lower reporter intensity relative to the control reporter in all stages of SGs (Figure 7F, G, J), indicating that the +39 site is required for upregulation of bam in SGs. In contrast, the -68 mutant reporter showed increased signal on SGs (Figure 7F, H, J), indicating that this site has an inhibitory effect on the bam promoter in SGs. This effect depends on the +39 site, as the increased signal was not observed when combined with the +39 mutation (Figure 7I, J). In contrast, activity of the +39 site does not depend on the -68 site, as the reduction of the signal did not change when combined with the -68 mutation (Figure 7I, J). These data suggest that the +39 site is used by the diffusible fraction of Dpp outside the niche to upregulate bam expression (Figure 7J, K).

It is still unclear how different concentrations of Mad can affect these two binding sites. Mad has been shown to interact with numerous co-factors to act as either a transcriptional repressor or activator (39). It is possible that Mad interacts stage-specifically with different cofactors on these sites. Moreover, Mad may indirectly regulate these sites through regulating other factors that bind to these sites. Further studies will be required to identify other factors involved in the opposing BMP signal output we observe.”

2. Discrepancies in Figure labels and annotations, missing description as well as multiple flaws in wording and syntax make the MS very hard to read and comprehend, especially for a non-specialist. Some of these points are listed under “minor remarks”, but this is just a very small fraction. The MS requires extensive and careful rewriting to reach the level required for publication.

We apologize these mistakes/mislabeled. We carefully went through and fixed them. Thank you.

Other points

1. In answering my question on whether the transgenic Dpp construct used to overcome embryonic lethality might somehow contribute Dpp ligands during spermatogenesis (thus complicating the analyses) they authors provide an additional experiment (described in line 169 and Fig S2H). Specifically, they show that depleting Dpp in the hub reduces pMad levels in GSC in the presence of the transgenic construct. In my understanding, unless the effect is absolute

(i.e. complete loss of pMad), this cannot disprove the contribution of ligands for the transgene. Similarly, since the authors agree that there are other sources of Dpp besides hub-derived Dpp, this experiment is not even adequate to define the hub as the main Dpp source.

As pointed out by this reviewer, we understand that this experiment does not prove there is absolutely no transgene expression in the testis. Therefore, it is possible that the diffusing fraction of Dpp is not completely depleted and perhaps only “reduced” in both morphotrap experiments.

However, our experiment (RNAi against mGL tagged Dpp) showed that depletion of mGL-tagged Dpp in the hub reduces pMad levels in the GSC, indicating a significant contribution of mGL-Dpp (but not the pPA dpp 8391/X transgene) in the niche signaling. This experiment proved that mGL tagged Dpp protein is functional, and that the phenotype we observed in the morphotrap experiment is surely representing loss of (or at least “reduction” of) Dpp function.

Moreover, to further ensure knock-down of all sources of Dpp, we have added data showing Dpp-Gal4(TS) mediated GFP (mGL-Dpp) knock-down in the presence of pPA dpp 8391/X and compared the results with Dpp-Gal4(TS)-mediated Dpp knock-down (new Figure S1A). We found that both showed similar levels of reduction of pMad in GSCs, indicating that these conditions both represent reduction of all sources of Dpp.

2. While the authors in the rebuttal letter agree that there are other sources of Dpp than the hub (see also above), the narrative of the papers still suggest exactly this. For example, the authors use the term long-range diffusion/distribution of Dpp, however since Dpp might come from cells adjacent to the differentiating cells (and not the hub), the diffusion might not be that long after all. Another example : even the schematic in Fig 2A suggest that the MTs trap Dpp that is derived from the same cells (hub) that also present the MT. I believe this needs to be clarified.

We have now revised these portions accordingly.

Minor suggestions

Line 83:

“hence we introduced a transgenic allele (pPA dpp 8391/X) containing genomic region of dpp expressed only in embryonic stage to assist embryonic development.”

Awkward phrasing.

Figure 1:

The figure has been “enriched” by a panel showing GFP-Dpp (panel F), however the panel (and the outcome of this experiment) are not explicitly discussed. Is this the allele from the Kornberg lab? If so (table 1 suggests that this is the only GFP-Dpp used), it seems that the distribution between cells is even better visible with this tool (compare 1E and 1F). Why not using this allele?

Also, while the table 1 is a good attempt to provide clarity over the available and used alleles, it would be good to use consistent nomenclature throughout the draft to make the life of readers (and reviewers) easy. For example, I cannot easily allocate the “ (Crispr knock in)” alleles of Suppl Fig. 1.

Line 112-118:

Fix syntax and grammar. All three sentences are flawed.

Line 149:

“Indeed, the Dpp-mGL signal in hub lysosomes after 4-hour chloroquine treatment showed increase in Dpp-mGL in hub cells of MT expressing testes!”

please use

Paragraph starting with line 152:

All figure/panel annotations are wrong, including the suppl. Figure. Fix!

“Indeed, the Dpp-mGL signal in hub lysosomes after 4-hour chloroquine treatment was increased in MT expressing testes”

Line 157:

“In comparison, mGL-Dpp, fasIII>Nrv-MT” replace with “In contrast, mGL-Dpp, fasIII>Nrv-MT”

Paragraph starting with line:

All figure/panel annotations are wrong, Fix!

Also legend for panels N,O misleading : N,O are not the of the same genotype as M, only O is (I hope)

Line 276:

“Next, we conducted trapped HA-tagged Dpp or Gbb using HA-trap” Unclear, do you mean

“Next, we trapped HA-tagged Dpp...” ? Would be also good to introduce the structure of the trapping system (as you did for MT)

We have now carefully examined the paper and revised these points, along with other textual errors, accordingly.

REVIEWERS' COMMENTS

Reviewer #1 (Remarks to the Author):

The authors have addressed all of the concerns that I raised in my previous review, and I find this draft suitable for publication. I am including a few minor suggestions below that would help add clarity to the final, published draft.

Line 97: the language should be change from "the rescuing transgene is unlikely to still be expressed" to "The rescuing transgene is not expressed in the niche to a level that is able to rescue defects resulting from knocking down mGL-Dpp."

Panel 1E: Add an arrow to emphasize the puncta distant from the niche to make this result stand out more clearly.

Line 111: Change "tags were inserted into upstream of last processing site" to "tags were inserted upstream of the last processing site"

Line 138: Change "descendant" to "descendants"

Figure 2A: Add a note to the legend explaining the BFP annotation on the schematic.

Line 160: Language "eliminated" should be changed to "dramatically reduced." I still see punctae, but I agree that they appear to be fewer and dimmer.

Line 201: "symmetric event" should be "symmetric division events"

Line 210: remove the underscore.

Line 219: Change "were" to "was"

Line 237: For clarity, change "supporting the idea" to "the data support the idea"

Line 266: Add "suggesting" just prior to the statement "that Tkv-mediated signaling pathway is also utilized" to soften the language.

Line 293: "pattern" should be "patterns"

Line 307-308: The authors discuss their data showing that trapping Gbb-HA in hub cells reduces pMad in GSCs. It would add clarity if they referred back to their similar data using the mcd8 MT with Dpp to trap Dpp at the niche.

Line 330: "triggers" should be "trigger"

Line 331: remove the word "specifically." Having this word adds confusion, since the authors just wrote about this model in another tissue (the wing disc)

Figure 6, Panels A',B',C', and D' would benefit from including the same dotted line labels shown in Panels A, B, C, and D, respectively.

Reviewer #3 (Remarks to the Author):

The authors now provide some new data and a partially modified MS that should address previous discrepancies.

While some of their arguments are valid, some other statements are - in my opinion unclear and some points have not been addressed. Here I focus only on the points that I have identified as major weaknesses in the previous rounds of review. In addition to these points, I still think that the MS requires solid proof reading.

1. Molecular analyses of bam transcription via bam reporters:

The authors decide to come back to date from the 1st MS version but removed from the 2nd version.

There is no evidence whatsoever that the -68 site is a Mad binding site !! The authors provide no explanation on how this site qualifies as a Mad/Smad binding site except for a very loose and short similarity to the previously (work on female bam expression) verified Mad site (+39). We are only left with a very vague, and completely unclear statement in the figure legend : "The previously two putative Mad binding elements are shown in boxes reported (38)" (I thought there has been a single, previously verified Mad binding element) referring to previous work with strong biochemical and reporter-based evidence on pMad binding to +39. In my opinion, and as stated in the first review, the second site could affect bam expression by any other means and not Mad binding. The regulatory sequence for the generation of the bam reporter is particularly short and I can easily envision how some nucleotide exchanges can cause slight quantitative changes in promoter performance. A causal connection between the element at -68 and pMad levels integration is lacking. The section also contains a new experiment (Fig. 7A,B) where ectopic BMP activation leads to groups of cells expressing bam reporters at different levels. The levels of bam reporter activation, the authors claim, correlate with the levels of pMad. Despite the quantification in 7B, this is difficult to correlate to the original data in 7A: The "low pMad group" contains cells that express the bam reporter at high, low or zero levels. In my opinion, this distribution might be arbitrary and not connected to pMad levels. Also the tumor cells of the "low pMad group" seem to accumulate pMad in the cytosol (or at least not in the nucleus), how is this consistent with the effects on bam expression?)

2. Dpp source

The authors acknowledged previously that the "diffusible" fraction of Dpp might be emanating from sources other than the niche and this has been reflected in some changed statements in the current version. For example, the concluding sentence of the first result chapter was changed from: "Taken together, these data indicate that Dpp can diffuse from the niche relatively long range (several hundred micrometers) through the extracellular space of the testis." to the awkward "Taken together, these data demonstrate that diffusible Dpp is freely diffusible in the testis." I guess they mean to say: Dpp is not only present in the niche but also detectable in the rest of the tissue. Still the narrative throughout the MS suggests that the low levels of Dpp found outside of the hub stem from Dpp produced in the hub (Example : line 426, "However, our work has demonstrated a diffusing population of Dpp away from the hub" or abstract). Thus, the authors favor a model where the Dpp forms a non-zero tail gradient in the tissue with all Dpp being produced in the hub. This is not supported by the data.

Some other points

The authors provide a long explanation in their response on how they exclude that the transgenic Dpp construct provides ligands in the testis. I did not get all the arguments but the explanation that "Dpp-Gal4(TS) mediated GFP (mGL-Dpp) knock-down in the presence of pPA dpp 8391/X and compared the results with Dpp-Gal4(TS)-mediated Dpp knock-down (new Figure S1A)" is a good point but there is no such experiment in Fig. S1A (unless I did not understand the experiment).

Below is the response for further reviewers' comments. Reviewer comments are blue, our responses are black.

Reviewer #1 (Remarks to the Author):

The authors have addressed all of the concerns that I raised in my previous review, and I find this draft suitable for publication. I am including a few minor suggestions below that would help add clarity to the final, published draft.

Line 97: the language should be change from “the rescuing transgene is unlikely to still be expressed” to “The rescuing transgene is not expressed in the niche to a level that is able to rescue defects resulting from knocking down mGL-Dpp.”

Panel 1E: Add an arrow to emphasize the puncta distant from the niche to make this result stand out more clearly.

Line 111: Change “tags were inserted into upstream of last processing site” to “tags were inserted upstream of the last processing site”

Line 138: Change “descendant” to “descendants”

Figure 2A: Add a note to the legend explaining the BFP annotation on the schematic.

Line 160: Language “eliminated” should be changed to “dramatically reduced.” I still see punctae, but I agree that they appear to be fewer and dimmer.

Line 201: “symmetric event” should be “symmetric division events”

Line 210: remove the underscore.

Line 219: Change “were” to “was”

Line 237: For clarity, change “supporting the idea” to “the data support the idea”

Line 266: Add “suggesting” just prior to the statement “that Tkv-mediated signaling pathway is also utilized” to soften the language.

Line 293: “pattern” should be “patterns”

Line 307-308: The authors discuss their data showing that trapping Gbb-HA in hub cells reduces pMad in GSCs. It would add clarity if they referred back to their similar data using the mcd8 MT with Dpp to trap Dpp at the niche.

Line 330: "triggers" should be "trigger"

Line 331: remove the word "specifically." Having this word adds confusion, since the authors just wrote about this model in another tissue (the wing disc)

Figure 6, Panels A', B', C', and D' would benefit from including the same dotted line labels shown in Panels A, B, C, and D, respectively.

We appreciate again for this reviewer for carefully reading our previous versions of manuscript and suggestions. We revised these points except of "symmetric division events" to fit our definition made in Figure 3A.

Reviewer #3 (Remarks to the Author):

The authors now provide some new data and a partially modified MS that should address previous discrepancies.

While some of their arguments are valid, some other statements are - in my opinion unclear and some points have not been addressed. Here I focus only on the points that I have identified as major weaknesses in the previous rounds of review. In addition to these points, I still think that the MS requires solid proof reading.

1. Molecular analyses of bam transcription via bam reporters:

The authors decide to come back to date from the 1st MS version but removed from the 2nd version. There is no evidence whatsoever that the -68 site is a Mad binding site !! The authors provide no explanation on how this site qualifies as a Mad/Smad binding site except for a very loose and short similarity to the previously (work on female bam expression) verified Mad site (+39). We are only left with a very vague, and completely unclear statement in the figure legend : "The previously two putative Mad binding elements are shown in boxes reported (38)" (I thought there has been a single, previously verified Mad binding element) referring to previous work with strong biochemical and reporter-based evidence on pMad binding to +39. In my opinion, and as stated in the first review, the second site could affect bam expression by any other means and not Mad binding. The regulatory sequence for the generation of the bam reporter is particularly short and I can easily envision how some nucleotide exchanges can cause slight quantitative changes in promoter performance. A causal connection between the element at -68 and pMad levels integration is lacking. The section also contains a new experiment (Fig. 7A,B) where ectopic BMP activation leads to groups of cells expressing bam reporters at different levels. The levels of bam reporter activation, the authors claim, correlate with the levels of pMad. Despite the quantification in 7B, this is difficult to correlate to the original data in 7A: The "low pMad group" contains cells that express the bam reporter at high, low or zero levels. In my opinion, this distribution might be arbitrary and not connected to

pMad levels. Also the tumor cells of the "low pMad group" seem to accumulate pMad in the cytosol (or at least not in the nucleus), how is this consistent with the effects on bam expression?)

We understand that -68 Mad site has never been tested for actual Mad binding. Although we are currently investigating interplay of Mad and other transcription factors on Bam promoter, this would be out of scope of current manuscript. Therefore, we added "putative" for -68 Mad binding site throughout the text as recommended by editor.

2. Dpp source

The authors acknowledged previously that the "diffusible" fraction of Dpp might be emanating from sources other than the niche and this has been reflected in some changed statements in the current version. For example, the concluding sentence of the first result chapter was changed from: "Taken together, these data indicate that Dpp can diffuse from the niche relatively long range (several hundred micrometers) through the extracellular space of the testis." to the awkward "Taken together, these data demonstrate that diffusible Dpp is freely diffusible in the testis." I guess they mean to say: Dpp is not only present in the niche but also detectable in the rest of the tissue. Still the narrative throughout the MS suggests that the low levels of Dpp found outside of the hub stem from Dpp produced in the hub (Example : line 426, "However, our work has demonstrated a diffusing population of Dpp away from the hub" or abstract). Thus, the authors favor a model where the Dpp forms a non-zero tail gradient in the tissue with all Dpp being produced in the hub. This is not supported by the data.

Thank you for finding these portions and the awkward description. We carefully went through and revised all misleading descriptions about origin of Dpp, including the Abstract.

Some other points

The authors provide a long explanation in their response on how they exclude that the transgenic Dpp construct provides ligands in the testis. I did not get all the arguments but the explanation that "Dpp-Gal4(TS) mediated GFP (mGL-Dpp) knock-down in the presence of pPA dpp 8391/X and compared the results with Dpp-Gal4(TS)-mediated Dpp knock-down (new Figure S1A)" is a good point but there is no such experiment in Fig. S1A (unless I did not understand the experiment).

The relevant data is presented in FigureS1A. 4th column is the data to show pMad intensity after Dpp-Gal4(TS)-mediated Dpp mGL knock-down (see below).

FigS1